# The invasion phenotypes of glioblastoma depend on plastic and reprogrammable cell states

Milena Doroszko[1,5], Rebecka Stockgard[1,5], Irem Uppman ®[1], Josephine Heinold[1], Faidra Voukelatou ®[1,2], Hitesh Bhagavanbhai Mangukiya ®[1], Thomas O. Millner[3], Madeleine Skeppås ®[1], Mar Ballester Bravo[1], Ramy Elgendy ®[1], Maria Berglund[1], Ludmila Elfineh[1], Cecilia Krona[1], Soumi Kundu ®[1], Katarzyna Koltowska ®[2], Silvia Marino ®[3], Ida Larsson ®[1,4] & Sven Nelander ®[1] ✉

Glioblastoma (GBM) is the most common primary brain cancer. It causes death mainly by local invasion via several routes, including infiltration of white matter tracts and penetration of perivascular spaces. However, the pathways that mediate these invasion routes are only partly known. Here, we conduct an integrative study to identify cell states and central drivers of route-specific invasion in GBM. Combining single-cell profiling and spatial protein detection in patient-derived xenograft models and clinical tumor samples, we demonstrate a close association between the differentiation state of GBM cells and their choice of invasion route. Computational modeling identifies *ANXA1* as a driver of perivascular involvement in GBM cells with mesenchymal differentiation and the transcription factors *RFX4* and *HOPX* as orchestrators of growth and differentiation in diffusely invading GBM cells. Ablation of these targets in tumor cells alters their invasion route, redistributes the cell states, and extends survival in xenografted mice. Our results define a close association between GBM cell differentiation states and invasion routes, identify functional biomarkers of route-specific invasion, and point toward targeted modulation of specific invasive cell states as a therapeutic strategy in GBM.

Glioblastoma (GBM), the most common primary brain cancer in adults, is characterized by rapid progression and a lack of effective therapeutic options for patients with recurrent disease. Unlike other difficult forms of cancer, GBM causes death not by distant metastasis but by rapid local invasion. The recurrence of GBM is attributed to infiltrative cells found in perivascular spaces, white matter, or brain parenchyma, also known as Secondary Scherer structures[1,2]. The amount of infiltration is negatively correlated with overall survival and tumor growth rate, as supported by surgical[3], radiological[4], mathematical[5],

and animal model studies. Yet, infiltrating cells are largely out of reach for current therapy. Comparisons between present-day patients and historical cases suggest that while the severe mass effect appears to be less common in GBM patients today, dissemination, including life-threatening brainstem invasion, is now more pronounced[6].

These observations raise several pertinent questions regarding GBM invasion. Specifically, is the observed impact of invasion on survival driven by particular subpopulations of invading cells? What cell-intrinsic and extrinsic factors mediate these invasions, and do they

[1]Department of Immunology, Genetics and Pathology, Uppsala University, Program for Neurooncology and Neurodegeneration, Uppsala, Sweden. [2]Department of Immunology, Genetics and Pathology, Uppsala University, Beijer Gene and Neuro Laboratory, Uppsala, Sweden. [3]Brain Tumour Research Centre, Blizard Institute, Faculty of Medicine and Dentistry, Queen Mary University of London, London, UK. [4]Department of Pediatric Oncology, Dana-Farber Cancer Institute, Boston, MA, USA. [5]These authors contributed equally: Milena Doroszko, Rebecka Stockgard. ✉e-mail: sven.nelander@igp.uu.se

vary among patients? Importantly, can targeting these invading cells or mitigating invasion extend survival in recurrent GBM? Recent molecular studies, including single-cell profiling, have identified transcriptionally distinct GBM subpopulations, influenced by both genetic mutations and the microenvironment[7–15]. Notably, GBM cells exhibit four main states: mesenchymal-like (MES-like), oligodendrocyte precursor cell (OPC)-like, neural progenitor cell (NPC)-like, and astrocyte (AC)-like[8]. The mesenchymal state, associated with increased invasion, has been found to rise over time in recurrent tumors[16]. Interestingly, Venkataramani et al.[17] reported that OPC/NPC-like states are prominent in invasion in vivo. Various pathways, including Eph- and epidermal growth factor receptor signaling, stemness pathways, and transcription factors like *SOX10* and *CEBPB*, have been linked to GBM invasion[14,18–22]. However, the genetic regulation and therapeutic targeting of GBM invasion remain largely unresolved.

Here, we investigate the hypothesis that GBM cell invasion routes are closely tied to their transcriptional states. Specifically, we aim to delineate which cell states favor perivascular versus diffuse invasion, identify key functional properties of these states, and pinpoint genes essential for each invasion type. By utilizing patient-derived cell culture xenograft (PDCX) models with diverse invasion patterns, we integrate single-cell transcriptomics and spatial proteomics to uncover distinct migration behaviors of GBM cell subpopulations.

## Results

### HGCC xenografts display a wide range of growth structures and invasion routes

The Human Glioblastoma Cell Culture (HGCC) Resource consists of extensively studied patient-derived cell (PDC) cultures, thoroughly investigated at genomic and pharmacological levels[23,24]. In our ongoing research, we have been systematically characterizing the invasion phenotypes of 64 GFP/luciferase-tagged HGCC cultures in nude mice, assessing the extent of perivascular and diffuse invasion, along with other morphological characteristics. The two predominant phenotypes identified in these studies (based on Principal Component Analysis) are either a consolidated tumor with perivascular invasion or a diffuse growth pattern, frequently involving the corpus callosum[25]. In order to investigate different modes of invasion of GBM cells, we picked six representative HGCC cell cultures with either predominant perivascular invasion or diffuse growth pattern, as suggested from their location in the PCA of phenotypic profiles of all cases in HGCC collection. We evaluated the growth structures using multiplexed immunofluorescence staining, employing STEM121 to identify tumor cells, and specific markers such as CD31 for blood vessels, MBP for white matter, AQP4 for astrocytes, and NeuN for neurons. Three of the chosen cultures (U3013MG, U3054MG, and U3220MG) produced bulky tumors with dense perivascular growth (Fig. 1A), whereas the

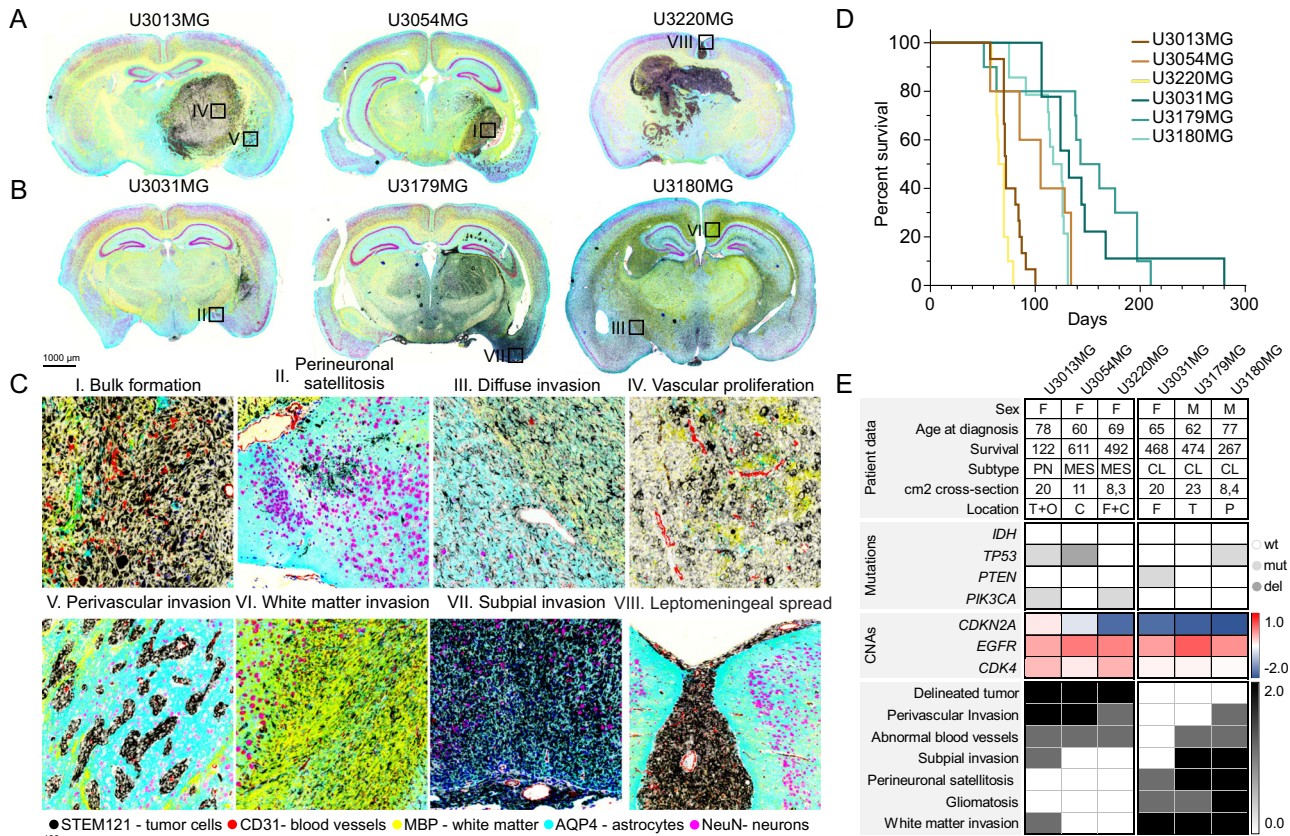

**Fig. 1 | Glioblastoma xenografts recapitulate known growth patterns. A** Coronal sections of mouse brains xenotransplanted with U3013MG, U3045MG, and U3220MG. **B** U3031MG, U3179MG, and U3180MG PDCs. Tumor cells are visualized in black (STEM121), blood vessels in red (CD31), white matter in yellow (MPB), astrocytes in cyan (AQP4), and neurons in magenta (NeuN). The scale bar indicates 1000 μm. Black squares indicate the zoomed-in location in (**C**). In both (**A** and **B**), representative scans were selected from 10 mice injected with each cell line. (I) Bulk formation, (II) perineuronal satellitosis, (III) diffuse infiltration, (IV) vascular proliferation, (V) Perivascular invasion route, (VI) White matter invasion route, (VII) Subpial invasion route, (VIII) Leptomeningeal spread. The scale bar indicates 100 μm. **D** Mouse survival in days for each injected patient-derived cell line. (*n* = 15 mice (U3013MG), *n* = 10 mice (U3054MG, U3220MG, U3031MG, U3179MG); and, 14 mice (U3180MG), **E** Table detailing patient data such as sex (F Female, M Male), age, survival (shown in days), subtype (MES Mesenchymal, PN Proneural, CL Classical), cm2 cross-section referring to the tumor volume, tumor location (T Temporal lobe, O Occipital lobe, C Cortex, F Frontal lobe, P Parietal lobe) mutational profiles, CNAs, observed phenotype, and invasion route. Source data are provided as a source data file.

other three (U3031MG, U3179MG, and U3180MG) produced a diffuse infiltration phenotype (Fig. 1B). Several different secondary Scherer structures were evident in our models (Fig. 1C), including leptomeningeal spread (U3220MG) and perineuronal satellitosis (U3031MG and U3179MG). Of note, the phenotypes demonstrated high reproducibility among mouse individuals (Supplementary Figs. 1 and 2), with concordance levels of 96% for diffuse infiltration, 88% for perivascular invasion, and 96% for perineuronal invasion (Supplementary Data 1). Interestingly, mouse survival rates varied between cases, with diffusely invading HGCC cultures showing a tendency toward longer survival times compared to those with bulk and perivascular growth and invasion phenotypes (Fig. 1D, logrank test: $\chi^2 = 9.08$, $df = 1$, $p = 0.0026$, $n = 45$ mice). The selected cultures had a spectrum of characteristic GBM mutations (Fig. 1E). In conclusion, these selected xenografts serve as representative examples of GBM with specific invasion routes.

## Transcriptional states define invasion routes in GBM xenograft models

Next, we aimed to elucidate the connection between the cell state distribution and the invasion phenotype in our PDCX models, utilizing single-cell RNA sequencing (scRNA-seq) for each culture. This encompassed samples from adherent cultures before injection and tumor cells isolated from mouse brains at experimental endpoint. The final data contained 119,766 cell transcriptomes, covering the six lines under in vitro and in vivo conditions, i.e., 12 groups (samples specified in "Methods"). The UMAP dimensionality reduction (Fig. 2A, B) and gene set enrichment of markers obtained by graph-based cell clustering (Fig. 2C) revealed distinct regions within the gene expression space for cells derived from the two classes of PDCXs. Specifically, PDCX models with bulk-forming and perivascular invading tumors populated a transcriptional subspace enriched for injury response and macrophage-like expression signatures, while diffusely growing PDCX models occupied a region enriched for neurodevelopmental, neuronal-like signatures. Oligodendrocyte-like signatures were observed for both invasion routes. Notably, the diffusely growing models were also enriched for outer radial glial cell markers and astrocytic markers (Fig. 2C)[26]. PDCX and PDC cells grouped together with cell cycle-related programs in a UMAP dimensionality reduction, confirming that all PDCX and PDC include cycling and non-cycling cells (Fig. 2A, C). Notably, U3220MG, which displays a high degree of leptomeningeal invasion (Fig. 1A, C), also harbored a distinct transcriptional cluster (Fig. 2A), suggestive of a unique cell state potentially linked to this invasion route.

Interestingly, the cells transplanted into mice showed a wider variety of cell states compared to those cultured in vitro (Fig. 2B, E and Supplementary Fig. 3). A possible explanation for this is that exposure to the mouse brain environment activates latent differentiation potential of the cells, whereas the cells stay less differentiated in vitro, which is maintained in stem cell conditions. We further computed cell state plots (cf. ref. 8), showing that the perivascular invading cultures showed a strong bias towards OPC-like and MES-like states, whereas the diffusely invading cultures were associated with NPC-like and AC-like states (Pearson's chi-squared test: $p < 2.22 \times 10^{-16}$, $df = 6$, Fig. 2D, F). This is intriguing since a previous characterization of invasive GBM, which focused on electrophysiological connectivity of the cells, found an important separation between unconnected NPC-like and OPC-like cells on the one hand, and connected AC-like and MES-like cells on the other hand[17]. This finding suggests that the preference for perivascular vs. diffuse invasion routes is orthogonal to the electrophysiological phenotype concerning cell state.

Taken together, we found a clear correlation between the invasion patterns of PDCX models and the unique cellular states they exhibited. Notably, perivascular invasion was marked by an abundance of OPC-like and MES-like states, while diffuse invasion was characterized by an NPC-like and AC-like state dominance. Of note, while this key difference was more evident in cells sampled from mouse brains, it was also

seen before injection, underscoring that the tendency towards a particular invasion phenotype and cell state distribution are intrinsic properties of GBM PDCs.

## Data-driven modeling reveals potential regulators of GBM invasion routes

Our initial scRNA-seq analysis revealed a significant correlation between transcriptional states and in vivo invasion routes (Fig. 2F). Subsequently, we employed a data-driven approach to identify potential regulators of GBM invasion.

We have previously described a method, termed single-cell regulatory-driven clustering (scregclust), to simultaneously cluster genes into modules and predict regulators (such as transcription factors and kinases) of these gene modules[27]. Applying scregclust to the scRNA-seq data from our PDCX and PDC models resulted in a regulatory landscape, where the different gene modules cluster based on their association with predicted upstream regulators (Fig. 3A and Supplementary Data 2). We assessed the modules by quantifying their similarity with established gene signatures of transcriptional states from ref. 8, as well as signatures of diffuse, perivascular, and leptomeningeal invasion routes fitted from our data (Fig. 3B). Upon inspection of the landscape, it became evident that groups of modules—referred to as metamodules—emerged across different PDCX models, displaying shared functional profiles and regulation (Supplementary Fig. 4A). By projecting the metamodule gene signatures onto a single cell atlas of human cortical development[28], we could also classify them according to their resemblance to normal cell types in the human brain, e.g., oligodendrocytes or astrocytes (Supplementary Fig. 4B). As a positive control of our regulatory predictions, we confirmed modules corresponding to the cell cycle programs G1/S and G2/M, predicted to be regulated by known cell cycle markers such as *E2F1* and *TK1* (G1/S), and *AURKA/B* (G2/M).

We used one-way ANOVA tests to analyze how the regulators of gene modules were influenced by factors such as growth conditions (in vitro vs. in vivo) (Fig. 3C), patient source (Fig. 3D), or invasion routes (Fig. 3E). Subsequently, we conducted a differential gene expression analysis between the perivascular and diffuse invasion routes to identify genes strongly associated with each route (Fig. 3F).

Our analysis identified a total of 53 regulators associated with invasion routes: 36 linked to growth condition or source patient, and an additional 17 with differential expression between perivascular and diffuse invasion (padj < 0.01, and differential expression log2 fold change > 0.5). Key regulators attributed to the perivascular route were *ANXA1* and *ANXA2*, two members of the annexin family, that play roles in inflammation and apoptosis[29]. Regulators for the diffuse invasion route included *HOPX*, *CKB*, *RFX4*, and *OLIG1*. *HOPX*, a homeodomain-containing transcription factor, is involved in stem cell maintenance[30]. *CKB*, an enzyme, regulates cellular energy homeostasis and is linked to cancer[31]. *RFX4*, a transcription factor recently identified as sufficient for the directed differentiation of CNS cell types from embryonic stem cells[32], *OLIG1*, essential for oligodendrocyte differentiation, contributes to central nervous system myelination[33]. Finally, for the leptomeningeal invasion route, *HMGA1* and *PRRX1* appeared as selective regulators. *HMGA1*, a chromatin-binding protein, is implicated in transcriptional regulation and cancer progression[34]. *PRRX1*, a transcription factor, contributes to embryonic development and cancer invasiveness[35]. *IFI16* and *HBGEF* are growth condition selective regulators, suggesting that these genes might be explored as markers of GBM cells responding to the tumor microenvironment in future independent work. Intriguingly, the transcription factor *MITF* and some of its known targets (*DCT*, *MLANA*, *PLT1*, and *S100A1*) - genes implicated in melanogenesis—were detected as a module active in bulk and perivascular invading cells. Moreover, *JUND*, *PDGFRA*, and *SCX* exhibited a high degree of patient selectivity, suggesting that these genes might have applications as robust biomarkers of inter-tumoral variation.

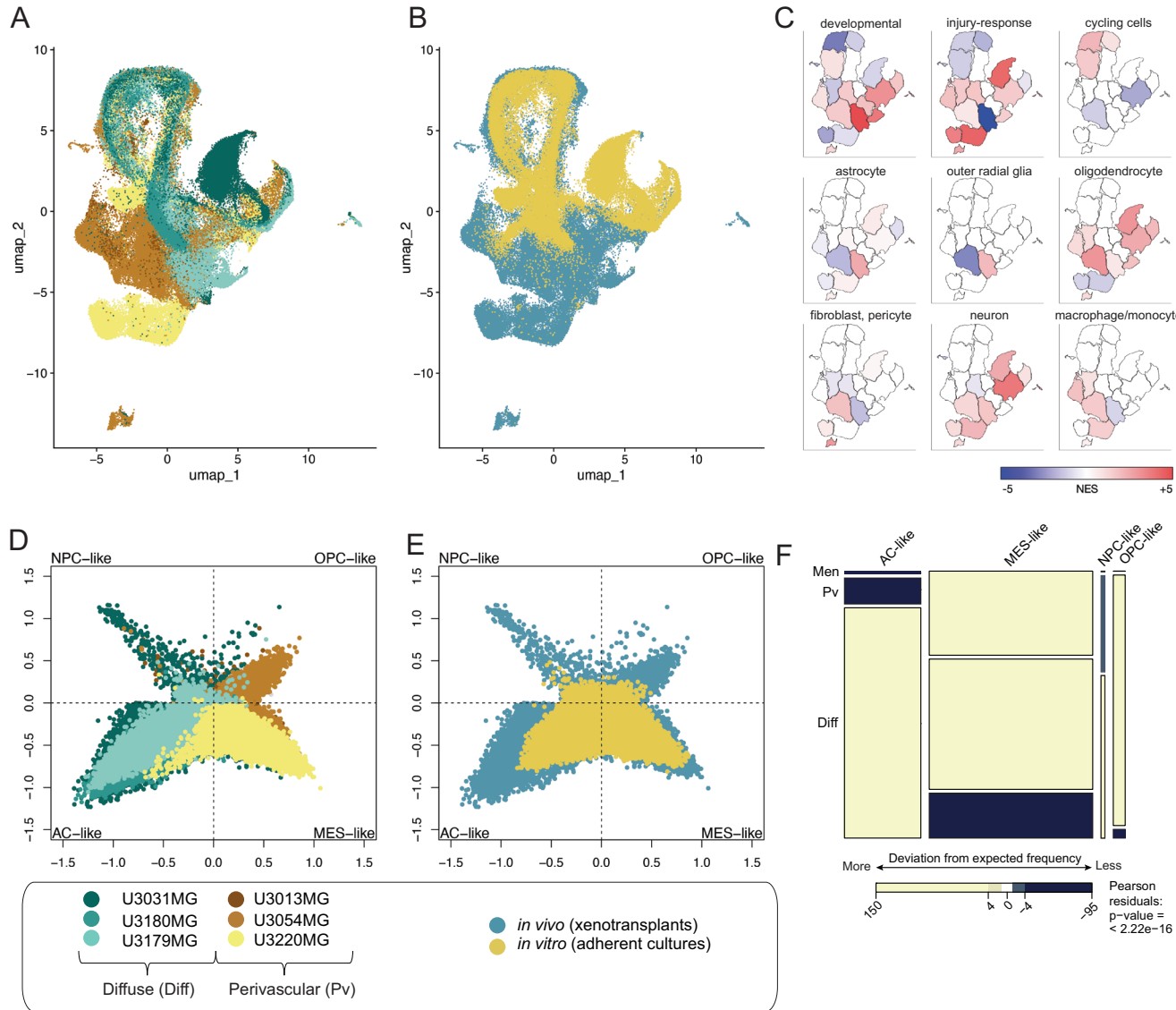

**Fig. 2 | GBM cells with distinct invasion phenotypes occupy distinct transcriptional states. A** UMAP separation of GBM cells by source patient suggests separation by invasion phenotype. (scRNAseq runs of $n = 1$ sample of in vitro cultured cells, $n = 2$ independent samples of in vivo PDCX-isolated tumor cells (from different mouse individuals) for each of the 6 GBM lines, except U3031MG, and U3179MG, which were run as $n = 1$ sample of in vitro cultured cells, $n = 1$ in vivo sample of PDCX-derived cells. The full data comprises a total of 119,766 single-cell transcriptomes and all cells are shown). **B** UMAP of the same GBM cells by growth condition, note that PDCX-derived cells occupy a greater set of transcriptional states. Same number of samples ($n$) and cells plotted as in (**A**). **C** UMAP of the same GBM cells as in (**A**, **B**), displaying enrichment of different gene signatures, measured by the Normalized Enrichment Score (NES) in each cell cluster. Note the differential distribution of injury response, oligodendrocyte, and macrophage signatures, versus neurodevelopmental signatures. Same number of samples ($n$) and cells plotted as in (**A**). **D, E** 4-state embedding (cf. Neftel et al.) shows MES/OPC enrichment of perivascular invading cells and NPC/AC enrichment of diffusely growing GBM cells. Same number of samples ($n$) and cells plotted as in (**A**). **F** Mosaic plot quantifying the relationship between transcriptional cell state and preferred invasion route, with coloring indicating observed frequencies compared to expected. A Pearson's chi-squared test of independence was performed ($p < 2.22 \times 10^{-16}$, $df = 6$, two-sided), with standardized residuals used for shading. No adjustments were made for multiple comparisons. Same number of samples ($n$) and cells plotted as in (**A**). Darker shades indicate greater deviation from expected frequencies.

In summary, Scregclust identified a concise set of genes with a possible upstream role in determining cell states associated with GBM invasion. To substantiate our findings, we compiled a shorter list of promising regulators to move forward with and validate at the protein level, as discussed next.

**Multispectral protein detection confirms markers of GBM route-specific invasion**

Our next objective was to validate the candidate regulator genes at the protein level by assessing their expression in different regions of the mouse GBM xenografts. For this, we combined 6-plex multi immuno-fluorescence staining with computational image segmentation to measure the expression of each protein in different spatial contexts (Fig. 4A, Supplementary Data 3, and Supplementary Fig. 5). To identify such contexts, we co-stained each protein of interest with markers for tumor cells (anti-human STEM121/NCL), blood vessels (CD31), and white matter (MBP). Utilizing morphological criteria and image k-means clustering, we segmented each slide into 9 different spatial compartments: high-density tumor (1), medium-density tumor (2), low-density tumor (3), circle-shaped aggregates (4), tumor cells growing in close proximity to vasculature (5), diffusely growing cells in the corpus callosum (6), diffusely-growing elongated tumor cells (7), mouse endothelial cells (8) and the mouse brain parenchyma (9) (Fig. 4B). As positive controls, we observed that CD31 produced a

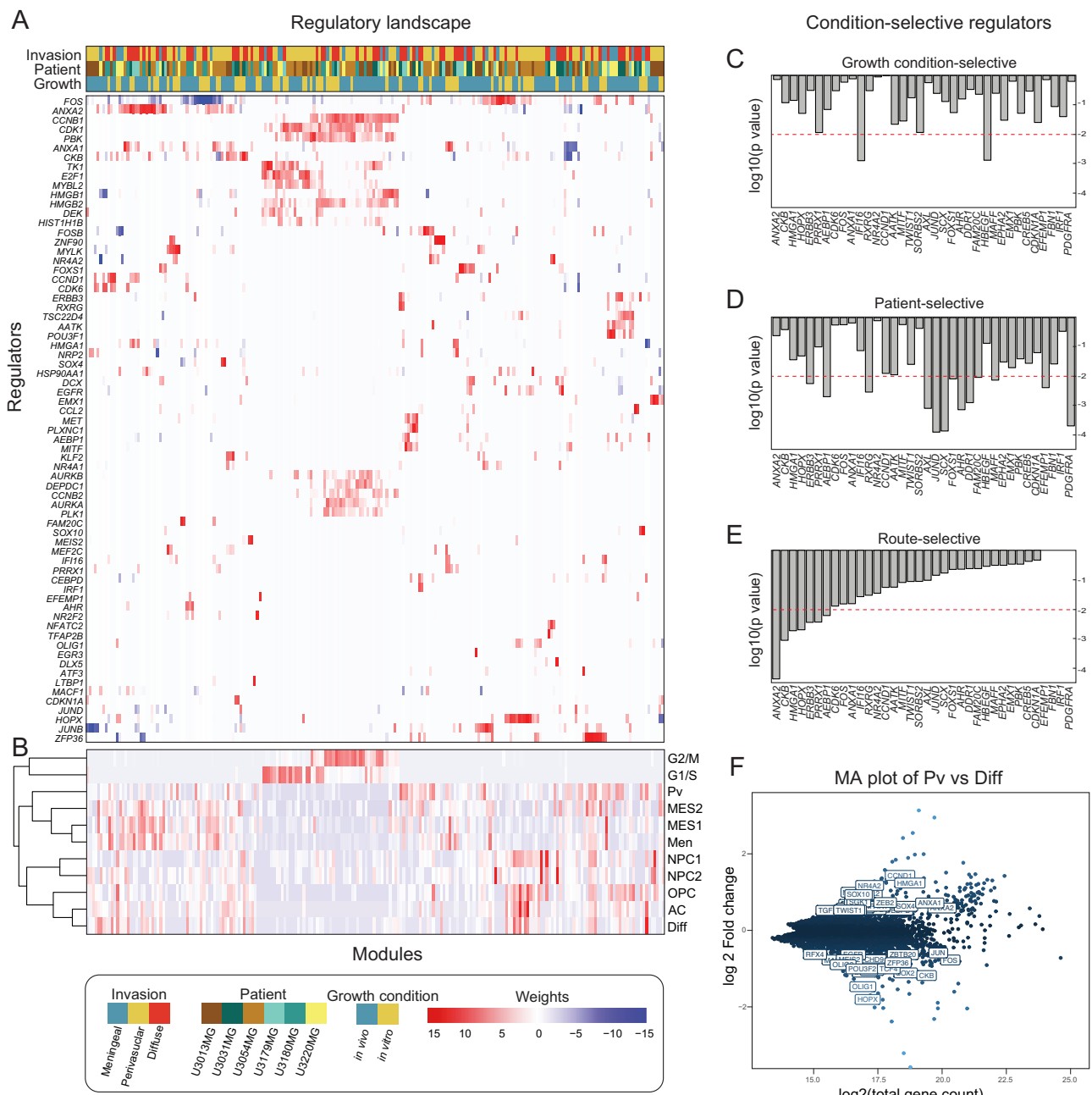

**Fig. 3 | The regulatory landscape of glioblastoma cells reveals potential candidates that regulate invasion routes. A** Heatmap of the regulatory landscape, with rows representing regulators (transcription factors and kinases) and columns representing gene modules, indicating sample origin. **B** Overlap between module gene content and cell state or invasion route signatures. **C–E** Barplots displaying regulators selective for growth condition, patient, and invasion route. One-way ANOVA tests were used to assess the effects of growth condition, patient origin, and invasion route. All ANOVA tests were two-sided with no corrections applied for multiple comparisons. **F** MA plot of differentially expressed genes, with labeled regulators from (**A**) with an absolute log2 fold greater than 0.5. Pv stands for perivascular invasion and Diff stands for diffuse invasion. The underlying data in (**A–F**) comprises scRNAseq runs of $n = 1$ sample of in vitro cultured cells, $n = 2$ independent samples of in vivo PDCX-isolated tumor cells (from different mouse individuals) for each of the 6 GBM lines, except U3031MG, and U3179MG, which were run as $n = 1$ sample of in vitro cultured cells, $n = 1$ in vivo sample of PDCX-derived cells. The full data comprises a total of 119,766 single-cell transcriptomes) Source data are provided as a source data file.

selective signal in the vascular spatial compartment (number 8), whereas STEM121/NCL was selective for all tumor-containing spatial compartments (Fig. 4C). Furthermore, in support of our computational segmentation, we confirmed that it accurately scored the relative abundance of different spatial compartments (e.g., the amount of dense tumor or perivascular cells), consistent with the manually observed phenotype of each PDCX model, grouping the cultures into dense/perivascular and diffuse clusters, respectively (Fig. 4D and Supplementary Fig. 6).

Analysis of all 6 PDCX models ($n = 240$ scans) showed that perivascular invading GBM cells exhibited higher expression of ANXA1 and CAV1 protein in their perivascular compartments (numbers 4 and 5) compared to diffusely invading cells (Fig. 4E). In line with the neurodevelopmental transcriptional phenotype observed for the diffusely invading cell lines, they displayed a relatively higher abundance of RFX4, AQP4, and HOPX (number 7). Notably, OLIG2 protein was enriched in elongated cells in sparse areas of the tumor, identifying it as a marker of cells that individually penetrate the brain parenchyma (Fig. 4C).

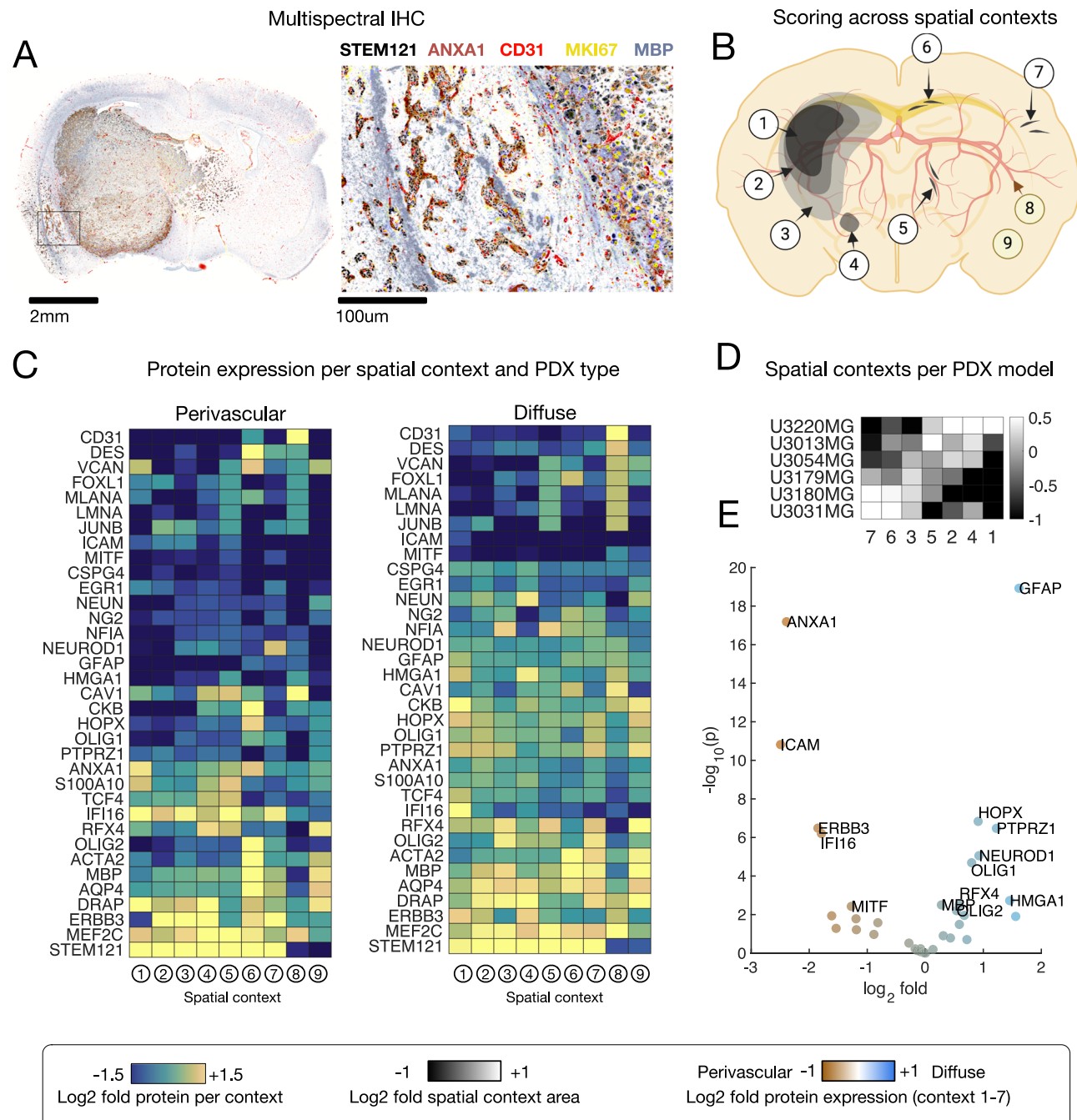

**Fig. 4 | Spatial proteomics reveals route-specific GBM invasion markers.**
**A** Multispectral IHC of U3054MG PDCX, with example staining of STEM121 in black, ANXA1 in brown, CD31 in red, MKI67 in yellow, and MBP in blue. Representative section from a total of $n = 10$ independent mouse replicates injected with U3054MG. **B** We segmented scans into 9 compartments (high-density tumor (1), medium-density tumor (2), low-density tumor (3), circle-shaped aggregates (4), tumor cells growing within close proximity to the vasculature (5), diffusely-growing elongated tumor cells in the corpus callosum (6), other diffusely invading cells (7), blood vessels (8), and mouse brain parenchyma (9). Created in BioRender. Nelander, S. (2025) https://BioRender.com/lpyogrt. **C** Scoring all PDCX models using 35 antibodies; upregulated and downregulated expression of proteins in named compartments for perivascular and diffusely invading cells. (Sections from $n = 2$ independent biological replicates (individual mice) were stained for each antibody, for each of the 6 cell lines). **D** Relative area of segmented compartments per PDCX cell line ($n = 3$ independent biological replicates (individual mice). **E** Volcano plot indicating key differentially expressed proteins. The log10 $p$-values are obtained from a two-sided heteroscedastic $t$-test, not adjusted for multiple comparisons (Sections from $n = 2$ independent biological replicates (individual mice) were stained for each antibody, for each of the 6 cell lines). Source data are provided as a source data file.

These findings underscore the heterogeneity of protein expression in GBM and further support ANXA1 protein as a marker associated with perivascular localization and dense growth patterns, and HOPX and RFX4 as candidate protein markers for diffuse route-invading GBM.

## Validation of *ANXA1*, *HOPX*, and *RFX4* as biomarkers of GBM invasion in patient samples

To assess the translational value of our laboratory findings, we investigated potential regulator expressions in human tissue microarray (TMA) samples from the HGCC biobank ($n = 148$) (Supplementary

Fig. 7 and Supplementary Data 6). Given the strong correlation of invasion routes with *ANXA1*, *HOPX*, and *RFX4*, these markers were chosen. Samples showed high expression of *ANXA1* in cells surrounding blood vessels, whereas cells with *HOPX* expression were found away from the blood vessels, which is in accordance with our PDCX data (Fig. 5A). *RFX4* expression was present in both normal brain tissue

and the tumor core area. Next, we asked whether these markers were associated with patient survival. Cox regression analysis (or multivariate survival analysis) with age, sex, and subtype as covariates revealed that ANXA1 protein expression (observed as the fraction of ANXA1-positive cells) had a slight association with worse survival (HR = 1.011, 95% confidence interval = [1.003,1.020], *p* = 0.00802),

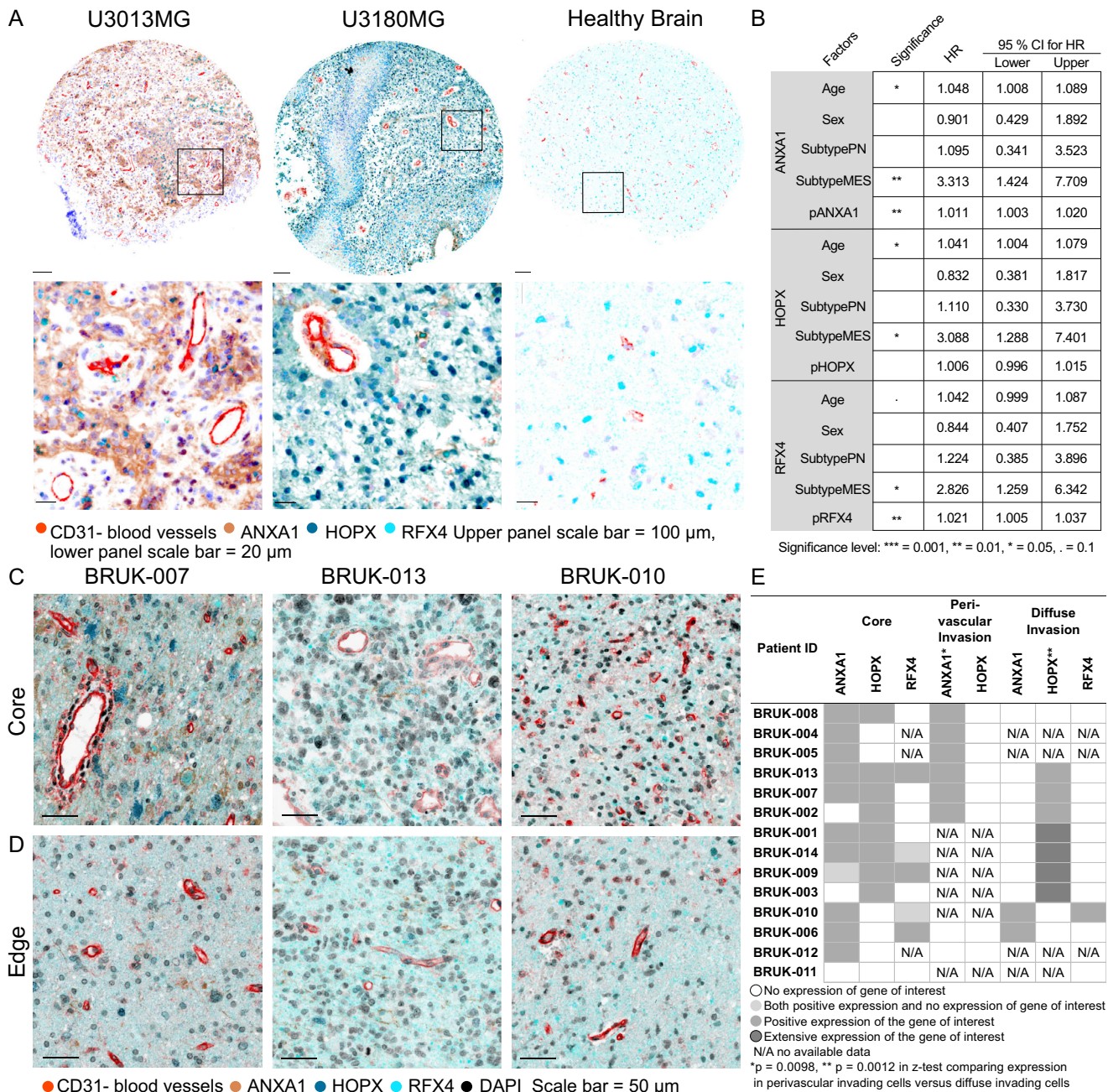

**Fig. 5 | Validation of route-specific invasion markers in an independent patient cohort. A** Human tissue microarray (TMA) staining of the tumor core, including patients U3013MG, U3180MG, and healthy brain tissue from HGCC. Staining includes CD31 in red, ANXA1 in brown, HOPX in blue, and RFX4 in cyan. The upper panel scale bar indicates 100 μm, while the lower panel scale bar is 20 μm. The stainings were repeated twice. Representative images chosen from *n* = 4 TMA cores from each patient. **B** Multivariate survival analysis using Cox regression on survival data from the HGCC, with age, sex, and transcriptional subtype as covariates, indicate associations between high ANXA1 protein (measured as the fraction of ANXA1-positive cells) and shorter survival, and between high RFX4 protein (measured as the fraction of RFX4-positive cells) and shorter survival. HR Hazard ratios, CI confidence intervals, and *p*-values are indicated in the figure. Two-sided test; no

corrections for multiple comparisons were made. **C** Staining of the tumor core and **D** edge from three patients from BrainUK. Staining includes CD31 in red, ANXA1 in brown, HOPX in blue, RFX4 in cyan, and DAPI in black. The scale bar indicates 50 μm. The stainings were performed once. Representative images from *n* = 1 tumor sample section from each patient, and 7 neuropathologist-inspected fields per section. **E** ANXA1 and HOPX proteins are selectively found in perivascular and diffuse regions in BrainUK samples, as determined by a two-sided *z*-test for proportions, assessing differences in marker expression between perivascular and diffusely invading GBM cells. Asterisks indicate statistical significance: *p* = 0.0098 (*), *p* = 0.0012 (**). *N* = 14 patients, all listed in the table. No corrections were applied for multiple comparisons. (N/A means that this type of invasion was absent in the sample). Source data are provided as a source data file.

while a high fraction of RFX4 protein positive cells was associated with worse survival (HR = 1.021, 95% confidence interval = [1.005,1.037], $p$ = 0.00856). No association between HOPX protein expression and survival was found (Fig. 5B). Extending the set of covariates further with individual key mutations (c.f. Fig. 1E) did not substantially affect these trends (Supplementary Data 4).

Although extensive, the HGCC biobank consists of samples of mostly European ancestry patients from a single hospital, and the tumor samples are from an unannotated core region. Therefore, to avoid bias from a single cohort, we also investigated patient tumor samples from an independent cohort, the Queen Square/NHNN repository (ethical approval was obtained via BrainUK, ref:21/014). Also, in this cohort, ANXA1 expression was observed localized to tumor cells near blood vessels, both within the tumor core and outside the tumor bulk. Since HOPX is also expressed in normal brain tissue, the distinction of its expression in the tumor core or border region is more challenging. Nevertheless, HOPX was expressed in neurons and glial cells, reflecting our PDCX findings. RFX4 expression was found in normal healthy brain tissue and scattered in the tumor core in some patient cases (Fig. 5C, D). Importantly, the expressions of ANXA1 and HOPX were found to be invasion route specific and not patient-specific also within this cohort (Fig. 5E).

ANXA1 has been investigated before in different cancer types[29]. In gliomas, ANXA1 has been shown to play a role in glioma progression[36], to be present in the immune microenvironment and to be correlated with survival and metastasis potential[37]. Less, however, is known about this protein's role in perivascular invasion in GBM. HOPX plays a critical role during normal development and is strongly expressed in radial astrocyte stem cells[38] and outer radial glial-like cells[26]. RFX4 functions as a transcription factor and may serve as a potential marker of GBM stem cells[39], with increased expression observed in gliomas[40] and implicated in astrocyte differentiation in cell models[41,42]. Furthermore, it correlated with poor GBM prognosis[39]. Our confirmation of these markers in two independent patient sample cohorts underscores their value in delineating cell populations potentially driving distinct types of invasion in GBM.

## Mice xenotransplanted with *ANXA1*-KO U3013MG, *HOPX*-KO U3180MG, or *RFX4*-KO U3180MG show increased survival and exhibit a shift in invasion phenotype

Next, we sought to evaluate the impact of *ANXA1*, *HOPX*, and *RFX4* on GBM cell invasion and survival. *ANXA1*, the predicted regulator of perivascular invasion, and *HOPX* and *RFX4*, the predicted regulators of diffuse invasion, were knocked out (KO) with CRISPR/Cas9 in U3013MG and U3180MG, respectively. These two cell lines were chosen due to their capacity for lentiviral modification. Cells were transduced with scramble (SCR) guide RNAs as controls. The KO was confirmed by PCR and sequencing of the flanked region, and by loss of protein expression for the markers expressed in vitro. Cell identity was confirmed with STR profiling (Supplementary Figs. 8 and 9). Before injecting the cells into mice, we conducted proliferation and self-renewal assays in vitro to ensure that *ANXA1*-KO, *HOPX*-KO, and *RFX4*-KO cells exhibited no discernible advantages in growth or tumor-forming capabilities (Supplementary Fig. 10).

We orthotopically injected nude mice with *ANXA1*-KO U3013MG cells, *HOPX*-KO U3180MG cells, and *RFX4*-KO U3180MG cells, along with corresponding SCR control. We then assessed survival, pathology, and gene and protein expression changes.

Mice grafted with *ANXA1*-KO U3013MG cells showed significantly extended median survival time (Fig. 6A, *p*-value < 0.0001) as compared to SCR control. To further analyze the impact of *ANXA1*-KO, we evaluated the brain of the xenografted mice histologically. We observed that *ANXA1*-KO U3013MG tumors did not form a bulk tumor as SCR-U3013MG and U3013MG-WT did (Fig. 6D, E). Specifically, the high-density tumor (1), medium-density tumor (2) areas abundance was decreased in *ANXA1*-KO,

as well as a decrease of tumor cells growing within close proximity of the vasculature (5). *ANXA1*-KO cells had a higher tendency to grow as a low-density tumor (3), and their morphology shifted from cell aggregates (4) to more diffusely growing elongated tumor cells (7) (Fig. 6D, E, I). In summary, the absence of ANXA1 in tumor cells reduced tumor bulk formation and significantly reduced association with vascular structures, with tumor cells shifting toward a more diffusely infiltrative phenotype. We did not observe significant changes in the number of proliferating cells compared to SCR controls (Fig. 6L).

In the diffusely growing U3180MG xenografts, targeting of either *HOPX* or *RFX4* prolonged survival, decreased tumor cell density, and (in the case of *RFX4*) led to altered morphology of the tumor cells. The KO of *HOPX* in U3180MG increased median survival (Fig. 6B, *p*-value = 0.0002) and these tumors appeared less aggressive than the control, as judged by reduction of tumor density (Fig. 6J). We saw no obvious phenotypic change of the tumor cells, except a possible increase in individual tumor cells making contact with blood vessels (Fig. 6G, J). The KO of *RFX4* also increased median survival time significantly (Fig. 6B, *p*-value < 0.0001). The *RFX4*-KO PDCX showed a marked reduction of tumor cells density (Fig. 6H, K). Additionally, a notable number of invading cells, often seen in the striatum, had a stellate phenotype, reminiscent of lower grade glioma (Fig. 6H, M).

## Dynamic analyses support a key role for ANXA1 in perivascular invasion

To broaden our understanding of the dynamics underlying the invasion phenotypes, particularly in the case of *ANXA1*, we performed real-time analyses comparing GFP-tagged U3013MG cells with wild-type *ANXA1* versus *ANXA1*se knockout (KO) cells across three complementary experimental systems (co-culture, zebrafish xenografts, and mouse brain slice grafts). In mouse brain slice assays, time-lapse confocal microscopy revealed significantly reduced migration of *ANXA1*-KO U3013MG cells along blood vessels compared to wild-type U3013MG cells, as quantified by single-cell tracking analyses (Supplementary Fig. 11 and Supplementary Movie 1). Consistently, in zebrafish xenografts, *ANXA1*-KO cells exhibited a pronounced tendency toward diffuse dispersion, whereas wild-type cells preferentially co-localized with vessels and demonstrated collective migration along these structures (Supplementary Fig. 12 and Supplementary Movies 2, 3, and 4). Similar observations were made in the co-culture system, where *ANXA1*-KO cells displayed markedly reduced interaction and adhesion to endothelial vessels (Supplementary Fig. 13). To assess the role of *ANXA1* further, we overexpressed *ANXA1* in U3013-*ANXA1*-KO and U3180MG cells. In the co-culture system, we saw that the *ANXA1*-KO phenotype was recovered and vascular association restored (Supplementary Fig. 13). Furthermore, when we overexpressed *ANXA1* in U3180MG cells, we saw that the diffuse growth phenotype was abolished in zebrafish xenografts. Cells instead formed a bulk (Supplementary Fig. 12 and Supplementary Movie 5). Additionally, an in vitro collagen sphere invasion assay comparing *ANXA1*-wild-type and *ANXA1*-KO U3013MG lines supported *ANXA1*'s involvement in regulating invasive behaviors, with knockout cells exhibiting significantly reduced invasive potential (Supplementary Fig. 10). Taken together, these complementary dynamic analyses support a role for *ANXA1* in promoting dynamic tumor cell association with blood vessels.

## The shift in preferred invasion route is accompanied by changes in transcriptional cell state

To understand the mechanisms underlying the altered growth and invasion phenotypes following KO interventions, we performed single-cell profiling of *ANXA1*-KO, *RFX4*-KO, and *HOPX*-KO tumor cells extracted from mouse brains. Cells from the diffusely invading *ANXA1*-KO tumors exhibited a transition from the MES- and OPC-like states observed in control *ANXA1*-WT tumor cells, favoring NPC- and

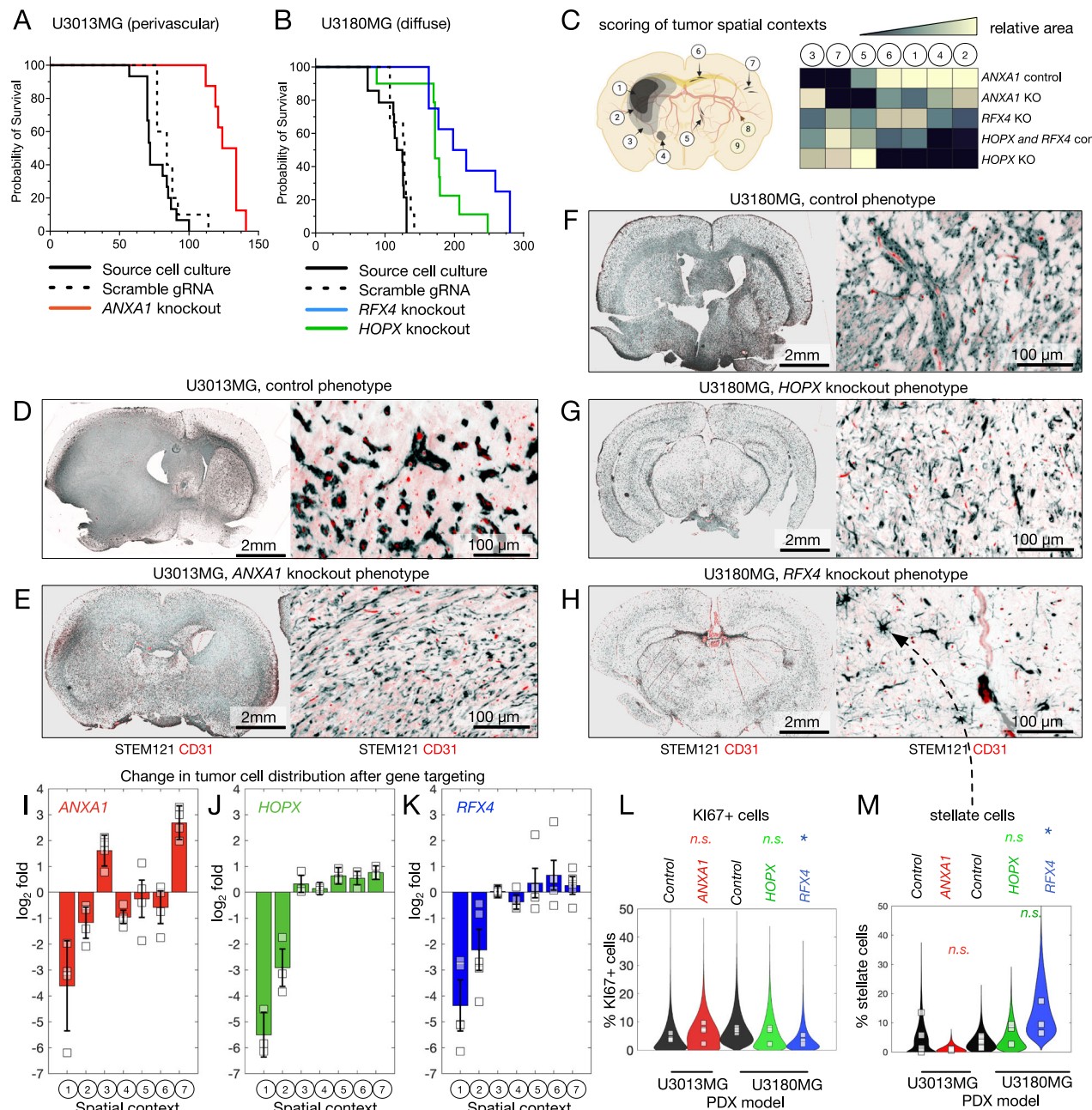

**Fig. 6 | Mouse xenotransplants of *ANXA1*-KO U3013MG, *HOPX*-KO U3180MG, and *RFX4*-KO U3180MG demonstrate prolonged survival and alteration in invasion phenotype. A**, **B** Mouse survival for *ANXA1*-KO U3013MG (*n* = 10 mice), *HOPX*-KO U3180MG (*n* = 10), and *RFX4*-KO U3180MG (*n* = 10). **C** Automated segmentation into 8 compartments. Created in BioRender. Nelander, S. (2025) https://BioRender.com/lpyogrt. **D**–**H** Whole brain scans and staining for each genotype. *n* = 3 brains were analysed per group and the stainings were repeated four times. **I**–**K** Change in compartment area for each PDCX-KO compared to SCR control. In (**I**, **J**), *N* = 4 independent replicate mice were used, and in (**J**), *N* = 3 independent replicate mice were used. Each mouse is shown as a point. Error bars are 90% confidence intervals obtained from a two-sided *t*-test, based on independent mouse replicates. **L** Percentage of KI67+ cells in each genotype (*n* = 4 independent

biological replicates (individual mice) in each group were used in the ANXA1 knockout vs control comparison, and *n* = 3 independent biological replicates (individual mice), were used in the RFX4 and HOPX knockout vs control comparison. Points represent individual mice, the distribution represents all counted fields in all mice. * indicates two-sided *t* test, *p* = 0.0375, calculated for the mouse independent replicates). **M** Percentage of stellate cells in each genotype. (*n* = 4 independent biological replicates, i.e., individual mice, in each group were used in the ANXA1 knockout vs control comparison, and *n* = 3 independent biological replicates, i.e., individual mice, were used in the RFX4 and HOPX knockout vs control comparison. Points represent individual mice, the distribution represents all counted fields in all mice; * indicates two-sided *t* test, *p* = 0.0139, calculated for the mouse independent replicates. Source data are provided as a source data file.

AC-like states (Fig. 7A, B). This trend toward astrocytic differentiation was further supported by differential expression analysis and gene set enrichment analysis (Fig. 7C). Additionally, we observed an upregulation of *GAP43*, a marker of regenerating neurons and reactive glial cells suggested to play a role in GBM invasion[43]. Anecdotally, the transcription factor *MITF* and some of its known targets (*DCT*, *MLANA*,

*PLT1*, and *S100A1*)—genes implicated in melanogenesis—were downregulated upon *ANXA1* loss (Supplementary Data 5 and Supplementary Fig. 14).

In contrast, knockout of *RFX4* in U3180MG xenografts significantly shifted cells toward NPC-like and OPC-like states, with gene signatures enriched for neuronal differentiation (Fig. 7D–F).

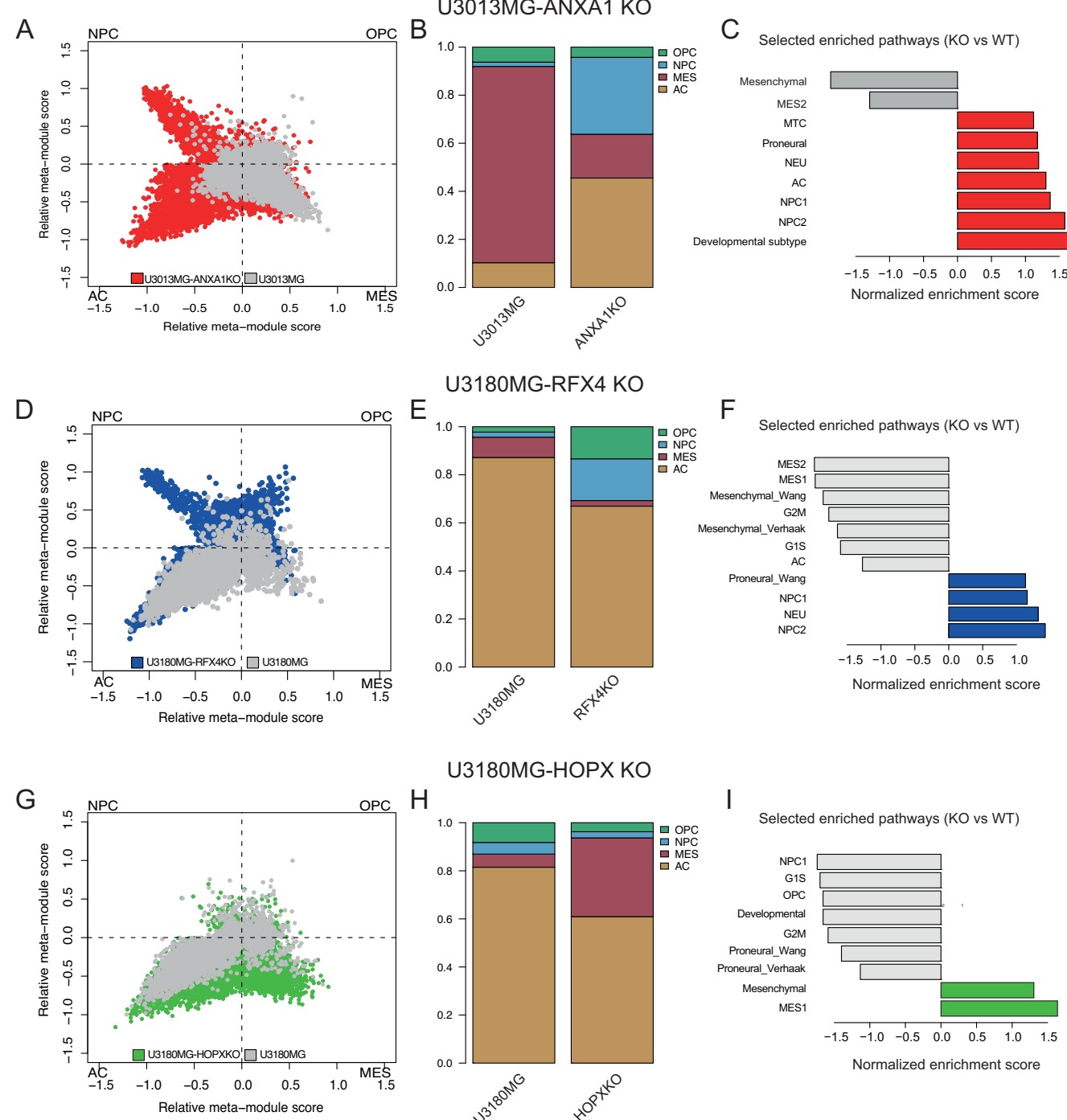

**Fig. 7 | Ablation of *ANXA1*, *RFX4*, and *HOPX* alter GBM cell state distribution and differentiation. A–C** Comparison of *ANXA1*-KO U3013MG cells with wild-type shows a shift towards NPC-like and AC-like differentiation. (scRNAseq data from *n* = 2 pools of tumor cells isolated from PDCX brains, total of 12069 cells). **D–F** Shift of cell state distribution in *RFX4*-KO U3180MG cells, towards an NPC-like, low-

proliferating state. (scRNAseq data from *n* = 2 pools of cells isolated from PDCX brains, total of 10824 cells). **G–I** Shift of cell state distribution in *HOPX*-KO U3180MG cells towards MES-like state. (scRNAseq data from *n* = 2 pools of cells isolated from PDCX brains, total of 13270 cells).

Additionally, *RFX4*-KO reduced the proportion of AC-like and MES-like states, alongside suppression of proliferation-related pathways. This decrease in proliferative cell populations corresponds with lower tumor density (Fig. 6K) and reduced KI67 positivity (Fig. 6L). Interestingly, *RFX4*-KO also resulted in decreased *HOPX* expression, suggesting regulatory interdependence between these transcription factors (Supplementary Fig. 14). The *HOPX* knockout in U3180MG cells prompted a notable transition toward MES-like states, accompanied by

decreased activation of developmental and proneural signatures (Fig. 7G–I). Although fewer cells adopted an AC-like phenotype after *HOPX*-KO, this shift was insufficient to fully eliminate astrocytic characteristics, potentially explaining the continued diffuse invasion behavior (Fig. 7G). The transition to a MES-like state was supported by increased vascular association observed histologically (Fig. 6G, J) and confirmed in co-culture assays showing enhanced endothelial interactions compared to controls (Supplementary Fig. 13). Together, these

findings establish *ANXA1* as a crucial factor maintaining MES-like states linked to perivascular invasion and identify *RFX4* and *HOPX* as critical regulators of proliferation and differentiation states in diffusely invading GBM cells. To further explore these regulatory dynamics, we compared knockout-induced transcriptional changes to an atlas of human early brain development by Eze et al.[44]. Mapping *ANXA1*-KO cells onto this developmental atlas revealed significant enrichment for radial glial-like phenotypes, marked by genes including *SOX9* and *PAX6*, with a concomitant gain of neuronal-like phenotypes. Cells with *RFX4*-KO predominantly exhibited upregulation of neuronal signatures, marked by e.g., *MAP2* and *TUBB3*. The results for *HOPX*-KO were more complex, highlighting enrichment for neuroepithelial clusters characterized by mesenchymal genes such as *ANXA2* and *ID3*, indicating a partial mesenchymal transition (Supplementary Fig. 15). In conclusion, knocking out *ANXA1* prompted GBM cells to adopt diffuse invasion accompanied by astrocytic differentiation. In contrast, while *RFX4* and *HOPX* knockouts also retained diffuse invasion, they distinctly altered the transcriptional landscape, affecting proliferation, differentiation, and mesenchymal traits. These findings highlight potential therapeutic implications, emphasizing the plasticity of MES-like states and the robustness of AC-like states, which could inform strategies targeting invasive GBM populations.

## Discussion

Extensive invasion, a hallmark characteristic of GBM, contributes to poor prognosis, patient mortality, and relapse. While various invasion routes exist, such as perivascular, diffuse infiltration, or perineuronal satellitosis, the underlying mechanisms remain elusive. Our results provide several pieces to the puzzle of brain tumor progression. Upon injecting patient-derived cells into the mouse brain, distinct known invasion patterns emerge that correlate with specific transcriptional states: MES-like cells exhibit perivascular invasion, while AC-like and NPC-like cells display diffuse invasion. Using a data-driven modeling strategy, we predicted possible regulators of these states, which were validated in patient samples, in vivo experiments, and in-depth molecular profiling.

In this study, *ANXA1* emerged as strongly associated with perivascular growth patterns in GBM. Knocking out *ANXA1* in perivascular invading cells induced notable phenotypic shifts, including the loss of tumor bulk and perivascular involvement, while acquiring an AC-like cell state and diffuse invasion, ultimately leading to increased median survival in mice. This observation suggests that the ANXA1+ perivascular invading phenotype potentially drives a reactive cell state, possibly linked to genes associated with injury response[45]. Furthermore, the over-expression of *ANXA1* in diffusely invading U3180MG cells caused bulk formation in zebrafish xenografts emphasizing the importance of *ANXA1* in forming bulky tumors. Mechanistically, ANXA1 plays roles in inflammation and tumor cell migration[46]. Cleavage of ANXA1 protein at the cell membrane generates a ligand for formyl peptide receptors, a class of G protein-coupled receptors involved in cell movement. Notably, targeting *ANXA1* increases radiosensitivity in GBM cell lines[36] and is expressed downstream of the ephrin B2 receptor (*EFBN2*), which is implicated in mouse (G26) models of perivascular invasion[14]. In our data, *EFBN2* was not selected by scregclust as a regulator because of its low expression in the human-derived PDCX models. The microenvironment, particularly the perivascular niche, significantly influences phenotypic expression, augmenting the perivascular invading phenotype and enriching proteins linked to mesenchymal transformation[47,48]. Our results appear consistent with the oncostreams phenotype[49], described as the collective invasion of COL1A1-positive tumor cells with mesenchymal properties. We propose that targeting ANXA1 may offer a strategy to suppress COL1A1-positive oncostreams in GBM, possibly with enhanced selectivity compared to targeting collagen 1 directly (since ANXA1 is less abundant in the normal brain than COL1A1). In epithelial cancers, epithelial-to-mesenchymal transition (EMT) has been linked to growth along blood vessels[50]. In accordance, we found intriguing overlaps between previously described regulators of EMT and genes detected in ANXA1-positive cells, including *TCF4*[47,48], and *S100A10*[51]. Future research endeavors include delineating regulatory dependencies and exploring the efficacy of targeting ANXA1 with small peptides[52] or exploiting it as a surface antigen for perivascular invading GBM cells. The association of NPC-like and AC-like cells to diffuse growth found in this work aligns well with the phenotype of non-malignant NPCs and astrocytes. Astrocytic and neural precursor migration is an integral part of brain development and injury response[53]. In response to injury, astrocytes transition from a quiescent to a migratory state, contributing to tissue repair and neuronal survival[45].

In contrast to the absolute loss of perivascular invasion and bulk formation upon *ANXA1* ablation, targeting the predicted diffuse invasion drivers *HOPX* and *RFX4* did not result in a complete loss of the phenotype in question, but rather a more complex shift in cell state linked to reduced proliferation and extended median survival in mice. The difference between the intervention experiments may point to the AC as a more robust cell state, consistent with what Schmitt et al. observed, that MES-like cells are more sensitive to reprogramming cues than other GBM states, which are more "hardwired"[54]. Knockout of *RFX4* suppressed AC-like transcriptional signatures and protein expression of GFAP, together with a higher expression of NPC-like signatures employing a more progenitor profile. *RFX4* drives the maturation of neural stem cells and neural structures[55,56], and our results point to a possible role in promoting AC-like states and growth in GBM. The presence of stellate cells in the *RFX4* knockout brains is intriguing and may point to a particular subpopulation that will require further investigation. Our knockout results point to *HOPX* as being downregulated upon *RFX4* targeting, potentially suggesting partially a shared mechanism between these two interventions on GBM invasion and growth. U3180 cells were pushed towards the MES-like state upon *HOPX* knockout. This was accompanied with an increase of vascular association in both in vitro (Supplementary Fig. 13) and in vivo (Fig. 6). When projected on the atlas of the developing brain, *HOPX* knockout cells were enriched for a neuroepithelial cluster (Supplementary Fig. 15). *HOPX* has been implicated in suppressing EMT in another cancer model before ref. 57. We have not investigated the exact mechanism underlying this mesenchymal shift however, all our findings support the connection between mesenchymal phenotypes and vessel association.

Each of the *ANXA1, RFX4,* and *HOPX*-KOs extended mouse survival times. We propose that the extended survival observed in mice grafted with *ANXA1*-KO cells is attributable to the absence of tumor bulk growth and perivascular invasion and the subsequent shift towards diffuse invasion. In these mice, the tumor cells appear more integrated into the brain tissue without forming a bulky mass that exerts pressure. As for the mice lacking HOPX and RFX4, although the exact mechanism is less clear, it's probable that the reduction in actively cycling cells contributes to their increased survival. While it's premature to extrapolate these findings directly to human patients, the correlation with improved survival outcomes suggests a potential clinical relevance. In the present cohort, we noted an association between RFX4 protein expression and shorter survival in unselected GBM patients, also after correcting for age, sex, and transcriptional subtype. RFX4 was also associated with survival within the subgroup of patients with a diffuse growth phenotype in mice (Supplementary Data 4). These findings may warrant validation in larger, independent patient cohorts.

Our results extend and complement previous studies aimed at relating cell differentiation to invasive growth in GBM. Firstly, Brooks et al. proposed a model wherein oligodendrocytic differentiation, dependent on *SOX10*, is observed among cells invading axonally in white matter tracts[21]. While the white matter invading phenotype was not a main focus of this study, our scregclust analysis detected a

cluster expressing oligodendrocytic markers, present in two of the cell cultures that we have characterized as bulky and perivascular. While oligodendrocytic protein markers were expressed in these PDCXs, we did not, however, see a specific expression of these markers in white matter-located cells (Supplementary Fig. 16). Secondly, Venkataramani et al. suggested that diffuse invasion is primarily driven by OPC-like and NPC-like cells[17]. Further examination revealed that a significant portion of diffusely invading unconnected cells consists of AC-like and NPC-like cells, supporting our observation that these cells utilize a diffuse invasion route. Lastly, Varn et al. identified two distinct GBM recurrence phenotypes: one characterized as neuronal and the other as mesenchymal, both linked with invasiveness[16]. This further supports both a mesenchymal mode of invasion and a neuronal mode of migration for the invasive GBM cells remaining in the normal brain parenchyma. Further work is needed to refine the nomenclature around cell states and invasion routes in GBM, and the association between AC-like cells and diffuse growth consistently observed across our three diffusely growing PDCX models extends previous observations. Towards this goal, studying GBM invasion across a larger clinical repertoire will be crucial. This would, for instance, open for robust statistical associations between tumor genetic and epigenomic features and their morphological presentation.

PDCX models must be used with an awareness of potential limitations. While the models recapitulate key invasion phenotypes observed in glioblastoma, they do not fully capture the clinical context of human disease. In particular, patients typically undergo surgical resection, radiotherapy, and mount adaptive immune responses against the tumor—factors absent in our immunocompromised mouse models. These differences likely contribute to the observed discrepancies in survival patterns between mice and patients, and we have interpreted our findings with these limitations in mind.

Methodologically, our study introduces a framework for uncovering invasive cell states and their regulators. In this study, we employ scregclust to identify key gene regulators implicated in perivascular or diffuse invasion, leveraging scRNA-seq data. Subsequently, we validate the protein expression of these regulators within the invasion niche of patient samples from two independent cohorts using multiplex immunofluorescence staining. Upon perturbation of a potential regulator in the invasion route of interest, we observe significant alterations in both RNA and protein expression, impacting the invasion route, migratory behavior, and morphology of these cells in vivo. Just like the observed transcriptional states are much more pronounced in the brain environment compared to adherent cultures, the effect of gene targeting is more pronounced in vivo than in vitro. It thus appears crucial to anchor the discovery of regulators of invasion in sufficiently complex models that recapitulate at least crucial parts of the brain environment. We acknowledge that our immunodeficient mouse models lack central aspects, which makes it important to validate the discovered functional biomarkers in independent patient materials, as was done here.

Taken together, this work presents a scalable approach to uncover critical genes that underlie specific cell states linked to brain tumor invasion. Looking ahead, it will be important to extend investigations to larger clinical repertoires, and to leverage our understanding of invasion regulators to interfere with these processes in a tailored manner. We reserve this for future work.

## Methods

### Patient samples
Patient-derived glioblastoma cell lines were established from tumor tissue as previously explained (ref. 23). All samples were collected with the informed consent of the patients, and the collection was approved by the Uppsala Regional Ethical Board, under number 2007/353. The cells were seeded on 1% laminin-coated flasks and maintained in serum-free neural stem cell medium with B-27 and N2 supplements, as well as EGF and FGF growth factors. The experiments adhered to the principles outlined in the WMA Declaration of Helsinki and the Department of Health and Human Services Belmont Report.

### Mouse xenografts: patient-derived xenograft model
All mouse experiments were conducted in strict accordance with an ethical permit granted by the Uppsala Animal Research Ethical Board, bearing reference numbers C41/14 and 5.8.18-06726/2020. Female NMRI nude (NMRI-Foxn1 nu/nu) mice were procured from Janvier Labs, while Hsd:Athymic nude-Foxn1 mice were procured from Envigo. Mice falling within the age range of 6 to 9 weeks were selected for the experiments. They were housed in individually ventilated cages, with each cage accommodating up to 5 mice. Appropriate housing enrichment, bedding material, food, and drinking water were provided ad libitum, and the mice were maintained on a 12/12-h light cycle. Human glioma cell cultures demonstrating verified tumor growth and the desired phenotype were systematically labeled with a lentivirus expressing GFP-Luciferase to enable subsequent tracking. All cell lines were STR profiled before injections to confirm their genetic identity (Eurofins Genomics). Orthotopic tumor injections were carried out by transplanting 100,000 labeled cells into the striatum of each mouse. The mice were monitored using in vivo bioluminescence imaging (luciferase monitoring) and regular weight measurements for up to 40 weeks post-injection. Humane endpoints were defined in accordance with approved animal ethics protocols and were used to minimize suffering. Mice were euthanized if they exhibited signs of significant weight loss (exceeding 15% of peak body weight maintained for more than one week), hunched posture, reduced activity (e.g., burrowing, social withdrawal), mild piloerection, or the onset of neurological symptoms such as incoordination or crouching. Additionally, body scoring was applied, with a termination threshold set on a predefined assessment scale. In cases where mice were monitored by luciferase imaging, elevated luminescent signal indicating progressive tumor growth also served as a humane endpoint, even in the absence of clinical symptoms. All mice were closely observed every 3–4 days, with additional veterinary consultation if the animals' conditions were unclear. Upon reaching the defined scientific or humane endpoints, mice were euthanized, and brains were harvested for histological or cellular analysis.

### Immunohistochemistry
For histology, mouse brains were processed in an automated tissue processor under the following conditions: 1 h 70% EtOH, 2 × 1 h 96% EtOH, 3 × 1 h 100% EtOH, 2 × 1 h Xylene, and 3 × 1 h Paraffin at 60 °C. The paraffin-embedded brains were then sectioned into 5 μm slides. Each block was analyzed for protein expression using a standard IHC protocol. In brief, after deparaffinization, antigen retrieval using Antigen Unmasking Solution Citrate-Based pH 6 (Vector Laboratories #H-3300) with 0.05% Tween 20 (Biorad #1610781) was commenced in 2100 Antigen Retriever for 15 min followed by cooling down to room temperature. Then sections were incubated in 3% H2O2 (30% H2O2 (Thermo Scientific #10687022) diluted in TBS) for 10 min, followed by washes with TBS-T (washing buffer). The sections were blocked with Normal Antibody Diluent (ImmunoLogic WellMed #UD09) for 30 min at room temperature, and primary antibodies STEM121 (1:500) (Takara #Y40410), ANXA1 (1:400) (CST, #32934S), RFX4 (1:500) (HPA #050527), and HOPX (1:1000) diluted in Normal Antibody Diluent were applied and incubated for 60 min at RT. We used BrightVision, a 1-step detection system, Goat Anti-Rabbit HRP (WellMed #DPVR110HRP), and anti-Mouse HRP (WellMed #DPVM110HRP) detection systems followed by incubation with Bright-DAB substrate kit (WellMed #KBS04-110). Slides were then counterstained with Myers' Hematoxylin and permanently mounted with Pertex (HistoLab #00811).

## Multiplex fluorescent staining and multispectral imaging

Multiplex staining was performed using the Opal 6-Plex Manual Detection kit (Akoya Biosciences, NEL861001KT). Procedures were conducted according to the protocol with a deviation, where anti-mouse HRP (Immunologic #DPVM110HRP) and anti-rabbit HRP (Immunologic #DPVR110HRP) were used for antibody detection instead. Antibodies were stripped after each Opal incubation using the microwave, and the procedure was repeated for the next primary antibody-Opal pairing. Every antibody-Opal pairing was independently validated as per the manufacturer's instructions. The validated antibody-Opal pairings are available in Supplementary Data 3. Slides were mounted with ProLong Diamond Antifade Mountant (Thermo-Fisher #P36970), imaged using the PhenoImager whole slide workflow, and unmixed using InForm 4.8 (Akoya Biosciences) software.

## Single cell isolation from PDCX tumors

Upon the experimental endpoint, mouse brains were harvested into cold HBSS buffer containing 1% Pen/Strep, 0.6% glucose, and 25 nM HEPES. Then, the brain was sliced using a 1 mm coronal section matrix, cut into about 1–2 mm pieces using a surgical blade, and dissociated into single cells using the Tumor Dissociation kit, human (Miltenyi, #130-095-929), used in combination with the Mouse Cell Depletion Kit (Miltenyi, #130-104-694) according to the manufacturer's protocols. Red blood cells were removed using the Red Blood Cell Lysis Solution (Miltenyi, #130-094-183).

## Single-cell RNA sequencing data generation

The generation of single-cell RNA sequencing libraries followed the manufacturer's guidelines, utilizing the Chromium Single Cell 3' Library and Gel Bead Kit v2, v3, and v3.1 (analysis of KO cells) (10× Genomics, Pleasanton, CA). Cryo-preserved cells underwent washing and re-suspension in 0.1% BSA in PBS just before loading onto a Chromium Single Cell B Chip (10× Genomics) with the aim of capturing 10,000 cells. Subsequently, the quality of the libraries was assessed using Agilent High Sensitivity DNA Kit and Agilent Bioanalyzer 2100 DNA Kit (Agilent Technologies). Libraries were sequenced on an Illumina NovaSeq 6000 with the sequencing configurations recommended by 10× Genomics. Demultiplexing, counting, and alignment to the human (GRCh38) reference genome were performed using Cell Ranger 3.0.2 (10× Genomics).

We profiled a total of 19 samples. Each of the cell lines U3013MG, U3180MG, and U3220MG were run as one in vitro sample, and two in vivo samples (from different mice). U3031MG, and U3179MG were run as a single in vitro sample. U3054MG was run as two replicate in vitro samples and four replicate in vivo samples (from different mice).

## Data processing, integration, and cell clustering

We performed single-cell analysis using the Seurat package (v. 4) (Butler et al., 2018). We filtered out cells expressing fewer than 500 genes and genes that were expressed by fewer than 10 cells. We filtered out potential doublets by setting nFeature_RNA parameters at greater than 7200 for v3 of the kit and greater than 5100 for the v2 kit. We removed low-quality cells that contained more than 30% mitochondrial genes, resulting in 110,458 cells retrieved (85.6% of the original population). We also removed highly expressed genes that are not related to the study, such as abundant ribosomal, mitochondrial, and hemoglobin genes. Lastly, to mitigate the effect of the cell cycle on cell groupings, we assigned each cell scores based on gene markers for the S- and G2/M-phases, and the difference between these scores was regressed out, as suggested by ref. 58. Then, we used the reciprocal PCA method to integrate the data and clustered cells using the Louvain algorithm with multilevel refinement. We used a range of resolutions from 0.01 to 1 to unravel cell subpopulations, and based on a directed graph calculated using the Clustree (v. 0.5.0) package to assess cluster separation, we

continued with resolution 0.3, which grouped the cells into 21 sub-populations. Note that batch integrated data was only used for visualization and clustering, not for downstream analyses described below.

## Regulatory landscape analysis by scregclust

The scregclust algorithm[27] was applied to the scRNA-seq data from each sample individually. For each run, the initial cluster number was set to 20, a minimum number of genes per cluster to 10, and a range of penalization values were tested (0.01, 0.05, and 0.1). The final penalization was chosen to 0.1 based on the metrics "predictive R2" and "regulator importance," as described in ref. 27. For each sample, this resulted in a regulatory table with regulators (transcription factors and kinases) as rows and gene modules as columns. The regulatory tables for all samples were merged into a common table for the entire sample set, and the data were z-transformed (Fig. 3A). Modules (columns) were clustered using hierarchical clustering, using the hclust-package in R with default settings (complete linkage) and Euclidean distance. Gene modules were characterized by quantifying their overlap with gene signatures of GBM cell states and cell cycle phases[8], as well as signatures representing the invasion routes (diffuse, perivascular, leptomeningeal). The overlap was quantified using the Jaccard index. Analysis of variance (ANOVA) tests were performed to assess the specificity of the predicted regulators in regard to growth condition, patient, and invasion route (Fig. 3B). Modules were given categorical annotations; the first two were derived from their sample origin (growth condition: in vitro/in vivo, and patient: U3013MG, U3031MG, etc.). The third, invasion route, was derived from the above-described scoring.

To compile the shorter list of regulators for experimental validation, we applied a combination of statistical and practical criteria. First, we prioritized genes that were significantly associated with invasion route (perivascular, diffuse, or leptomeningeal), based on ANOVA and differential expression analyses (adjusted $p$-value < 0.01, absolute log2 fold change > 0.5). We excluded regulators that were predominantly associated with patient identity or growth condition, as shown in Fig. 3C–E, to avoid confounding effects. From the resulting list, we selected regulators with established or plausible roles in invasion, differentiation, or transcriptional control, and for which high-quality antibodies were available for spatial protein validation. This process led us to focus on ANXA1, RFX4, and HOPX, which emerged as strong and feasible candidates.

## Metamodules

Metamodules were defined through hierarchical clustering of the merged regulatory table described above, and cutting the dendrogram at height 36, which resulted in 13 metamodules. Signatures were defined by, for each metamodule, merging the gene content of each individual gene module and keeping genes that were common for four gene modules or more. These metamodule signatures were then used to assign a metamodule score to each cell in the dataset provided by ref. 28, using the AddModuleScore()-function in the Seurat R-package.

## Spatial analysis of PDCX tumors

To score the intensity of different protein markers in different anatomical niches, we processed the VP images as follows. We regard each pixel as a point in 8-dimensional space ($z_1, z_2, \ldots, z_8$) where four of the channels were common to all analyzed images: nuclei (DAPI), auto-fluorescence (AF), tumor cells STEM121/NCL, and endothelial cells (CD31). The four remaining channels were used to evaluate proteins of interest. Images were loaded from qptiff format using bioformats toolbox in Matlab. For each channel, we used L2-regularized regression to correct for shared variation with the other channels. The L2 penalty was set to 0 for DAPI and AF channels, and to a tuning constant for the others. After correction, we segmented a four-channel image consisting of the DAPI, AF, STEM121/NCL and CD31 channels using image $k$ means segmentation (Matlab image analysis toolbox), with $k$ set to 5.

This consistently resulted in a 5-cluster solution with easily identifiable centroids representing tumor cells (high STEM121/NCL) and endothelial cells (high CD31). Pixels assigned to these centroids were used to obtain binary images T and V representing the tumor (T) and vascular (V) parts of the section. The endothelial niche was defined as the set of positive pixels in the V image. High, medium, and low-density regions of T were found as positive pixels of the T image in regions of different density, as measured by the Matlab imboxfilter method. We subsequently used a set of morphological property filters to detect elongated tumor cells, tumor cells near blood vessels, tumor cells near vasculature, and tumor cells in dense bundles. After these steps, we had obtained a labeling matrix L, that provided the class of each pixel. We subsequently scored each protein i by measuring its average intensity in each class j, correcting for cellularity using the DAPI channel, i.e.,

$$score(i,j) = \frac{1}{|S_j|} \sum_{(x,y) \in S_j} \frac{z_i(x,y)}{z_{DAPI}(x,y)}$$

where Sj is the set of pixels (x,y) in class j.

### Tissue microarray study from HGCC cohort and survival analysis
The stitched pyramidal OME-TIFF files were loaded into QuPath 4.3 software[59], and TMA was disarrayed to assign coordinates to the TMA cores. The nuclei were segmented using the StarDist 4.0 extension[60]. Then, for each protein marker, we evaluated staining specificity and set up manual classification thresholds depending on their localization. These fixed thresholds for each marker protein were then used for the classification of all TMAs within the set. The process was separately iterated for each staining set.

To assess whether marker expression was associated with patient survival, we conducted multivariate Cox proportional hazards regression using the R survival package (v3.4-0). The model incorporated the fraction of positively stained cells per core (i.e., positive cells/total nuclei) as the predictor, with age, sex, and transcriptional subtype as covariates. Multiple cores per patient were accounted for by clustering on patient ID, using the R syntax: coxph(Surv(time, status) p̃rotein + age + sex + subtype, data = dataset, cluster = patient_ID). Transcriptional subtype[12] was included as a categorical variable with three levels (classical, mesenchymal, proneural) with classical used as the baseline subtype by construction. We included subtype as a covariate due to its association with survival in univariate Cox regressions ($p < 0.05$). We also explored adding common mutations (c.f. Fig. 1E) as covariates, and performing the survival analysis in subsets of patients, defined by their mouse xenograft growth pattern (Supplementary Data 4). Mutation and mouse growth data was obtained from refs. 24,25 and phenotypic subsets found by 2-class k means clustering. All patients in this analysis were deceased at the time of data collection (i.e., no censored observations) and all cases were IDH wild-type.

### BrainUK cohort
To extend our collection and provide material that consisted of invasive regions of glioblastoma, we applied for access to samples from BrainUK (BRAIN UK Ref: 21/014). We then checked the expression of our top candidate proteins ANXA1, RFX4, and HOPX using mIF staining as described above. The material was carefully analyzed and scored by neuropathologists in 7–10 fields of view per section, selected in tumor-invaded niches.

### Lentiviral transduction
For generating knockout clones of target genes (*HOPX*, *ANXA1*, *RFX4*, and SCR) for U3013MG and U3180MG, cell cultures were transduced using a reverse-transduction method. Briefly, cells were detached using TrypLe, washed in PBS, and counted. Then, 100,000 cells were co-transduced with the Cas9-nickase and gRNA vectors. To minimize off-target effects, cells were transduced with the Cas9-nickase vector at MOI 3 and the gRNA vectors at MOI 5. After vector addition, cells were incubated for 2 h at 37°, then plated onto laminin-coated 6-well plates. The virus-containing medium was replaced after 24 h, and selection medium was added 3 days post-transduction. Cultures were treated with antibiotic selection medium for 7–10 days and then passaged for seeding each of them into 96-well plates as single-cell clones using FACS. See Supplementary Figs. 17 and 18 vectors and guideRNA sequences.

### Genotyping PCR and Sanger sequencing
Single-cell clones constituting colonies were genotyped to identify knockout clones. DNA was isolated using lysis buffer and incubated for 2 h at 60°. DNA was precipitated using precipitation buffer for 30 min at RT and washed 3 times with 70% EtOH. The pellet was air-dried for 30 min and then resuspended in TE buffer (pH 8). Clones with visible alterations in amplicon size from high-throughput PCR were selected for Sanger sequencing. In the second step, KAPA HiFi HotStart ReadyMix was used to amplify the DNA, and the amplicons were separated on a 2% agarose gel. The purified amplicons were then subjected to Sanger sequencing. Details of primers used are in Supplementary Fig. 19.

### Knockout evaluation
Sanger sequencing results were qualified and analyzed using SnapGene and the ICE CRISPR analysis tool. Clones with knockout indication were expanded, and about 10 million cells were collected from each clone to create FFPE cell pellets for IHC analysis of protein. A small pellet was also collected for second genotyping PCR and sent for Sanger sequencing. FFPE cell pellets were sectioned and stained with antibodies and protocol indicated in Section "Results". From clones with confirmed protein loss, a pellet of 100 thousand cells was collected and sent for STR profiling. Three to ten knockout clones per target were mixed in equivalent numbers 6 days before injection in mice.

### Proliferation, self-renewal, and invasion assays on knockout clones
To assess the proliferation and self-renewal capacities of knockout cells, we used CyQuant Cell Proliferation Assay and Extreme Limiting Dilution Assay (ELDA). In the proliferation test, cells were seeded in a range of serial dilutions in duplicates and allowed to grow for 72 h. After that, the Cyquant Protocol was performed according to the manufacturer's instructions. Self-renewal was tested by seeding cells in dilutions ranging from 200 to 1 cell per well in 96-well ultra-low attachment plates over the period of 7 days, two biological replicates were used. ELDA analysis was conducted using software accessible at http://bioinf.wehi.edu.au/software/elda/, following the specified procedure. Invasion was assessed by seeding 3000 cells/well to 96-well S-Bio plates and spheres were allowed to form for three days. After sphere formation, fresh media was added followed by addition of 1:1 Matrigel (Corning, #356234) and media mixture on top on ice. The plate was then transferred to 37° and followed up to 10 days. The invasion capacity was assessed using Incucyte software.

### Ex vivo PDCX brain slice culture assay
Live PDCX slice culture assay was performed as described most recently by us[61]. Briefly, GFP-tagged xenograft tumors from U3013MG-ANXA1 SCR and knockout lines were grown in nude mice. Brains with optimal luciferase signals were extracted and placed in ice-cold HBSS (Gibco, #24020117), then embedded in low-melting agarose-HBSS in square molds. Using a Leica VT 1200 S vibratome, 300 µm brain slices were cut at speeds of 20–200 µm/second and transferred onto transwell membranes in 12-well plates (Corning, #3460) with brain slice culture medium containing 2.5 mM HEPES, 10 mM glucose, and

2 µg/ml Tomato lectin-DyLight 594 for vasculature visualization. Excess medium around the slices was removed to maintain an air-liquid interface. Slices were incubated at 37°, 5% $CO_2$. The next day, plates were placed in the Image × press Micro Confocal system (Molecular Devices), capturing time-lapse images every <2 h for up to 5 days. Media was changed every 48 h. Images were processed, stitched, and overlaid in MetaXpress 6.5, and frame-to-frame registration was done using custom MATLAB scripts for analysis. Peritumoral regions of the slice were subjected to cell detection with image analysis operations for blob detection, followed by single-cell tracking with a Kalman-filter based framework written in MATLAB (Image Processing Toolbox, Computer Vision Toolbox, Version 23.2, Release 2023b, The MathWorks, Inc., Natick, Massachusetts, United States). A convolutional neural network was used to classify cells as vessel-associated or other. The proportion of cells in either class was calculated as the average proportions of classes from several peritumoral regions. Cell speed (microns per hour) for each cell was calculated as the average speed for the whole life-time of the cell track.

### In vitro 3D co-culture assay and analysis
Human brain microvascular endothelial cells (HBEC-5i) (ATCC, #CRL-3245™), at a density of 13,055 cells per well, were seeded on top of a Matrigel base in a µ-slide 15 well for 3D culture (Ibidi, #81506). The plate was incubated in a humidity chamber at 37° with 5% $CO_2$ for endothelial tube formation. The GBM PDCs were labeled with Qtracker™ 525 cell tracking dye (Invitrogen, #Q25041) by incubating 1 million cells in a 0.1 nM labeling solution for 1 h at 37°. Soon after the formation of vessel-like scaffolds by endothelial cells, GBM PDCs, at a density of 4100 cells per well, were seeded on top of the endothelial network. The plate was transferred to the Image × press Micro Confocal system for live imaging. Time-lapse Z-stack images (step size: 5 µM) were acquired at 15-min intervals for a maximum of 31 h and saved as 2D maximum projection images for cell migration and statistical analysis. Using a custom written analysis framework written in MATLAB (Image Processing Toolbox, Computer Vision Toolbox, Version 23.2, Release 2023b, The MathWorks, Inc., Natick, Massachusetts, United States), cell centers were identified and tracked throughout the time-lapse using a Kalman-filter based framework as described above. A binary mask based on the endothelial cells and enlarged with morphological dilation was used to categorize cells as being associated or non-associated with the vessel-like network. The proportion of cells associated with the network for each perturbation was estimated by sampling cells with replacement from replicates to obtain a distribution of values for statistical testing.

### Zebrafish xenograft generation, imaging, and analysis
An incross of Tg(kdrl:mCherry) labelling vasculature on pigmentless Casper strain (nacre−/, roy orbison−/−) background was generated to obtain embryos for tumor injections. The glioblastoma cells, expressing GFP and luciferase, were resuspended in NSC medium containing 20 mg/ml polyvinylpyrrolidone (PVP; Sigma #PVP360) and injected into zebrafish embryos at 1 day post fertilization (dpf). Stills were obtained by TCS SP8 DLS LightSheet microscope (Leica) and 24-h time-lapse imaging was performed with Confocal SP8 (Leica), both at 24 h post injection (hpi). By using surface rendering, 3D representations of the tumor cells and blood vessels were generated. Colocalization analysis was performed using Imaris (Bitplane v.9.5), by identifying the regions of colocalization between these rendered surfaces and calculating their proximal distance.

### Statistical analyses
*Cell-state plots.* Cell-state plots were generated as described. Barplots in Fig. 7B, D and F were generated by counting the number of cells in each quadrant. *Mosaic plot.* To statistically assess the relation between cell state and invasion route, a chi-square test was performed and visualized as a mosaic plot. *Differential gene expression analysis and MA plots.* Differential gene expression (DGE) analysis was performed using the FindAllMarkers-function in the Seurat package. MA plots were created by plotting the log2FC-values from the DGE analysis against the log2 total gene count for each gene across cells.

### Reporting summary
Further information on research design is available in the Nature Portfolio Reporting Summary linked to this article.

## Data availability
The single cell RNA sequencing data generated in this study have been deposited in GEO database under accession ID GSE270083. The source data is available in Supplementary figures and the source data file. Matlab code for this project is found on https://zenodo.org/records/15682177. Source data are provided with this paper.

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

## Acknowledgements

We thank the involved patients and their families for the support and donation of materials to the Human Glioblastoma Cell Culture (HGCC) biobank. We also thank the HGCC team for invaluable contributions in collecting and providing the patient-derived cell cultures used in this study. We thank the ongoing HGCC Tissue Microarray (Tobias Bergström, unpublished) and HGCC Phenobank initiative (Cecilia Krona, unpublished) for sharing TMAs and phenotypic data, respectively. We thank the Brain UK biobank for making patient materials investigated in

Fig. 5 available. We thank the National Genomics Infrastructure (NGI) for providing the sequencing service and the BioVis Platform for providing assistance with FACS sorting and microscopy. We thank FoUU for assistance with sectioning and scanning tissue slides and Artur Mezheyeuski for sharing expertise on multiplex staining, multispectral image acquisition, and data analysis. We thank Finn Hallböök, Karin Forsberg Nilsson, and Veronica Rendo for their valuable comments and feedback during the writing process. This research was supported by the Swedish Cancer Society (20 0839 PjF), the Swedish Research Council (2021-03224), Knut and Alice Wallenberg Foundation (2022-0057), and Swedish Foundation for Strategic Research (CCS23-011).

## Author contributions

Experiments were performed by M.D., R.S., I.U., J.H., F.V., H.B.M., M.B.B., L.E. and R.E. Mouse experiments were performed by S.K., R.S., I.U., M.B.B., M.B., M.D., H.B.M. and C.K. (coordination). Profiling and imaging data were analyzed by M.D., I.L., M.S. and S.N. Human tissue was analyzed by T.M. and S.M. Zebrafish experiments were analyzed by F.V. and K.K. The first manuscript draft was prepared by R.S., I.U., I.L., S.K., M.B.B., M.D. and S.N., with input from the other authors. S.N. guided the study.

## Funding

## Competing interests

The authors declare no competing interests.
