## [Transparent Peer Review file · Nature Communications]

The invasion phenotypes of glioblastoma depend on plastic and reprogrammable cell states

Corresponding Author: Professor Sven Nelander

Version 0:

Reviewer comments:

Reviewer #1

(Remarks to the Author)

In this study, Doroszko et al use PDX models and computational approaches to examine the potential links between tumor cell state and invasion routes. They show that cell lines enriched for MES-like and OPC-like state preferentially form dense tumours with perivascular invasion patterns, whereas lines enriched for NPC-like and AC-like fates are less dense and invade more diffusely. They then identify regulatory modules for each phenotype coalescing around ANXA1 for perivascularly invading tumors and HOPX/RFX4 for diffuse tumors. These are then functionally tested by CRIPR/Cas9 KO revealing a shift towards diffuse invasion in ANXA1-KO tumours and changes in fate, infiltration and proliferation in RFX4-KO tumors. The authors also show that KO of all three regulators extends survival.

This is overall an interesting paper, which is well written and executed. The computational approaches are a key strength of the study and reveal interesting links between cell fate and invasion patterns, that represent an important finding for the field. There are however some weaknesses, particularly in the functional studies, which need to be addressed in a revised manuscript to better support the authors conclusions.

1. It is not appropriate to refer to unpublished work in demonstrating the two invasion patterns. This is one of the main findings of this study and the data should be shown, including the survival curves to better support the correlation between invasion modes and disease severity.
2. While the role of ANXA1 in driving the formation of dense lesions is well supported by the data, its specific role in perivascular invasion is not. Its expression is broad, the differences in expression in perivascular cells are not clear from Figure 4C. More evidence to support a direct role in invasion is required, staining of 'static' terminal tissue is not sufficient. As it stands the observed effect could also be a byproduct of the dense, mesenchymal phenotype and not due to a regulatory role for ANXA1 of invasive behavior. The authors should assess invasion in real time in the presence or absence of ANXA1 (e.g. in ex vivo systems) and consider conditional approaches for deleting ANXA1 at later stages of tumorigenesis to disentangle regulation of dense growth from invasion.
3. To more convincingly demonstrate a role for ANXA1, gain-of-function experiments would also be required. Does expression of ANXA1 in diffuse lines switch them to dense/perivascular invading tumours?
4. The results of the HOPX and RFX4 KO experiments are puzzling and do not fully support a key role for these genes in diffuse infiltration. Their effects seem multifactorial precluding a clear interpretation of the data. The increase in perivascular invading cells in HOPX-KO is not convincing (Figure 6J). In addition, the reduced proliferation observed in both lines is a major confounder. The phenotypes observed may be simply due to reduced proliferation and altered differentiation and have little to do with invasion itself. Again, invasion should be assessed directly here and in real-time.
5. The GFAP results are also confusing as the authors show in Figure 7D/E that there is no change in the proportion of AC-like cells, yet they then show a decrease in AC-like signatures and GFAP staining in these tumours. Is AC fate really

changing and is GFAP a robust readout for this? In general, the experiments relating to HOPX and RFX4 inactivation are too underdeveloped in their current form.

Minor points

Sentence: "these latter were also enriched" on page 5 is confusing and should be rephrased.

It would be helpful to label the PDX models with their migration pattern in addition to the U number in the text and figures.

The clarity of Figure 2C could be improved.

The difference in survival shown in Figure 5b are not convincing. It would be more appropriate to move this panel to a supplementary figure.

Reviewer #2

(Remarks to the Author)

Doroszko et al. present an interesting paper on the invasion phenotypes of glioblastoma. I must confess this biological areas is quite distinct from my own but I am familiar with the computational and experimental techniques so I won't comment on the usefulness of the results to the community. I appreciate through the careful references to controls and potential confounding aspects of the analysis.

The approach looks sound and generally good care is taken throughout the manuscript to explain the intent and logical flow. The place where the manuscript could use most improving is the methods section where I am left guessing in many places what the others actually did. I suspect they are sensible things but I wouldn't be able to reproduce the analysis myself from this description. A list of uncertainties I had are below but I think the authors could apply a general principle of being more precise with the methodology descriptions:

- 1) Which highly expressed genes were removed from the analysis (currently vague)?
- 2) What is cell-cycle regression?
- 3) Louvain with multilevel refinement -> why was this chosen over other options?
- 4) Range of resolutions from 0.01 to 1 what was actual values tried?
- 5) How was the dendrogram computed (e.g. average linkage?)
- 6) "we assessed the stable and therefore optimal cluster separation" - There are two problems here I don't know what you actually did and I don't know how the second part of the sentence follows
- 7) Range of penalization value - again would be helpful to be precise
- 8) What was the actual penalization parameter chosen and what were the values of the metrics.
- 9) Hierarchical cluster settings missing again.

The figures were generally hard to read if they contained numeric or textual detail. Whilst many could use improvements those in figure 7 I could not really deduce what I was meant to read.

"Our initial scRNA-seq analysis revealed a significant correlation between ..." - at the moment this is simply a claim with no evidence, a figure or some data is missing I think.

Reviewer #3

(Remarks to the Author)

Doroszko et al. investigate the invasion phenotypes of glioblastoma (GBM) and how they are influenced by plastic and reprogrammable cell states. The study characterizes the invasion phenotypes of 64 GFP/luciferase-tagged HGCC cultures in nude mice (although this data is not provided), identifying two main phenotypes: consolidated tumors with perivascular invasion and diffuse growth patterns. Six cultures were selected for further analysis based on their consistent phenotypic expression. The authors use scRegClust to identify regulators associated with invasion routes, with specific genes linked to perivascular (e.g., ANXA1, ANXA2) and diffuse (e.g., HOPX, CKB) invasion. The findings suggest that specific transcriptional states correlate with distinct invasion patterns in GBM, with ANXA1 identified as a key regulator of perivascular invasion.

Major comments

1. The authors do not discuss the relevant clinical information from the patients and their consistency or disagreement with their experimental findings. For example:

a. Example 1: The survival patterns of both groups in host mice do not correlate or agree with the actual survival of the patients according to Fig 1 E and D. The entire purpose of modeling tumors in mice is to produce conclusions that help us predict the behavior of these tumors in human patients, but in this case, that isn't true and isn't discussed. There are many likely reasons for this, such as tumor-immune interactions. Since the mouse modeling was performed in immune-

compromised mice, lacking an essential component of the tumor microenvironment, conclusions from these experiments should be carefully evaluated and thoroughly discussed.

b. Example 2: Fig 1 E (patient clinical information) shows that all 3 PDCs with diffuse invasion phenotype are derived from patients with Classical subtype tumors, while the perivascular PDCs come from Mesenchymal and Proneural tumors. Previous reports (Nefel et al.) showed that while each patient tumor is composed of a mixture of cell states, proneural subtype tumors are mainly composed of NPC-like and OPC-like cells, Mesenchymal subtype tumors are mainly composed of Mes-like cells, and classical tumors contain more AC-like cells. While this somewhat agrees with Figure 2D,E, it was not discussed. Additionally, stacked bar plots showing the distribution of each subtype per sample would be a valuable addition to show the exact composition of each sample since butterfly plots can be tricky to interpret, especially when cells are close to the axis.

c. Are there any H&E from the original 64 (or at least the 6 used in the paper) patients provided by the HGCC that confirm this phenotype in humans?

2. The authors claim that the dimension reduction of the scRNAseq data revealed “distinct regions” for the cells derived from the two classes. However, the method used for integration (RPCA) has been benchmarked and shown to produce poor results. That being said, did the authors consider any other integration approaches? Was the sex of the patient (two males and four females) accounted for? Multiple batches? etc.

3. The authors claim that “U3220MG displayed a specific cluster apparently associated with its high degree of leptomenigeal invasion (Figure 2A).” The wording of this phrase is slightly confusing. How was the association determined? Was there any molecular/marker analysis done to back up this statement? Is this observation consistent through the 2 PDX samples and 1 tissue culture sample from this line?

4. Some experimental details are confusing/missing. For example, while the authors mention that they used 19 samples for the scRNAseq in Fig 2A, they do not mention how many samples per group (12 groups (6 cell lines and 2 conditions)). The deposited data in GEO show only 16 samples that are mentioned in this figure and 4 additional samples with ANXA1KO and RFX4KO (not 19).

5. How was the “shorter list of promising regulators” compiled? What were the cutoffs or criteria used?

6. Were multivariate survival analyses performed accounting for age/subtype/sex and mutations? Since these markers were associated with different growth phenotypes, which supposedly have different survival implications, it does not make sense that all markers predict worse survival in patients. Additionally since the growth patterns of these PDXs have been assessed, the survival should be conducted for each growth pattern separately.

Minor comments:

1- Figure 3 labels/gene names are too close to each other and hard to read

2- Legends are missing important information that is needed for the reader to understand the figures for examples:

a. Figure 1E, what is the unit of survival (days?). Also abbreviations/locations should be defined.

3- Some such information is present in the method but should be included in the legends for examples:

a. In Figure 5B, the type of “expression” should be clearly annotated protein IHC vs transcriptomics. The high/low threshold should be mentioned in the legends. It would also be beneficial if the expression scores per patient for all the markers assessed to be provided for all 148 patients alongside their clinical information. This information should be provided in the legends. This is especially important since the HGCC resource that’s available online only contains 48 samples as opposed to the 148 samples. The online resource provides gene expression profiling data that is downloadable in z score format. When using this information to assess survival, neither ANXA1, HOPX or RFX4. Although ANXA1 and HOPX show the same trends. It’s worth mentioning that in the TCGA GBM dataset neither of these markers are prognostic. When using the TCGA GBMLGG combined dataset, the markers are prognostic but it’s only suggestive of association with Glioblastoma not low-grade gliomas. This may be a distinction between protein vs RNA expression but needs to be discussed.

4- PDXs are serially transplanted from mouse to mouse and not cultured first. The point of PDXs is to mimic the in vivo component and since these cells were handled in vitro prior to transplantation (cultured, genetically engineered to express GFP-Luciferase), I believe they’re no longer true PDXs and the distinction should be made (Mouse xenografts from PDCs rather than PDXs).

5- Supplementary table with differential expression information between clusters/samples from the scRNAseq should be provided.

• Some experimental details are confusing/missing. For example, while the authors mention that they used 19 samples for the scRNAseq in Fig 2A, they do not mention how many samples per group (12 groups (6 cell lines and 2 conditions)). The deposited data in GEO show only 16 samples that are mentioned in this figure and 4 additional samples with ANXA1KO and RFX4KO (not 19).

Version 1:

Reviewer comments:

Reviewer #1

(Remarks to the Author)

The authors addressed most of my comments, particularly regarding the functional studies on ANXA1. The manuscript has improved greatly overall, though the mechanistic work on RFX4 and HOPX remains underdeveloped. It would be important to tone down the statements around ANXA1 playing a specific role in perivascular invasion in the final manuscript. The presented data only support a role for ANXA1 in vascular association and formation of a bulk, but no conclusive evidence on perivascular invasion is provided. The conclusions are therefore overstated in their current form.

Reviewer #2

(Remarks to the Author)

The reviews have responded well to by comments. I have no further comments.

Reviewer #3

(Remarks to the Author)

Overall, the authors have made significant efforts to address the reviewers' concerns and provide additional data to support their conclusions. The manuscript is now more robust and comprehensive, but some comments require further attention.

Points Not Addressed:

1. In my original comments (#6), I stated: "Were multivariate survival analyses performed accounting for age/subtype/sex and mutations? Since these markers were associated with different growth phenotypes, which supposedly have different survival implications, it does not make sense that all markers predict worse survival in patients. Additionally since the growth patterns of these PDXs have been assessed, the survival should be conducted for each growth pattern separately." I appreciate the additional analyses performed by the authors; however, the multivariate analyses only account for sex and age and not subtype and mutations. Additionally, an important point was not addressed. Since the growth patterns of at least 64 of these PDXs are assessed by the authors and available, why not perform the survival analysis separately by growth pattern?
2. In my original comments (#4), I stated: "Some experimental details are confusing/missing. For example, while the authors mention that they used 19 samples for the scRNAseq in Fig 2A, they do not mention how many samples per group (12 groups: 6 cell lines and 2 conditions). The deposited data in GEO show only 16 samples that are mentioned in this figure and 4 additional samples with ANXA1KO and RFX4KO (not 19)." While the authors kindly responded by adding the missing samples to the GEO repository, they did not clarify the statement in the manuscript regarding those 19 samples. These samples can be summarized into 12 groups (2 conditions and 6 lines) with some conditions having 2-4 replicates while others having a single replicate. I do not understand the need to explicitly state the number 19 without clarifying what is in those 19 samples. Was there a specific reason why 3054 PDXs required 4 replicates?
3. In the rebuttal, the authors' response to comment number 5 is not reflected in the edits to the referenced methods in section 8.
4. While the new legends include appropriate annotation of the expression used for survival (protein), the methods still state: "Survival analysis (multivariate). Cox proportional hazards regression was performed using median-dichotomized gene expression levels (e.g., HOPX, ANXA1, RFX4) as covariates alongside age and sex, with survival time as the outcome."
5. More importantly and adding to point #4. When I attempted to replicate the survival results, I was met with a couple of challenges. #1 survival status is not provided (1 or 0, I assumed that all are 1s for this test) #2 There are multiple points per patient in supplementary table #11 with no mention in the methods of how these replicates are accounted for. Specifically, 52 patients have 4 replicates, 8 patients have 3 replicates, 82 patients have 2 replicates, and 8 patients have a single replicate. I was able to generate an almost identical Kaplan Meier curve for RFX4 only when using all 394 data points with provided survival information but not when using a single (averaged) point per patient which is the correct analysis. This is extremely concerning and needs to be urgently addressed.

Minor Comments:

1. The legend for Fig 2C is a little confusing. It states: "UMAP of GBM cells displaying enrichment of different gene signatures of the clusters. NES refers to normalized enrichment score. **Perivascular invading cells (yellow/brown)** exhibit enrichment for injury response, oligodendrocyte, and macrophage signatures, while diffusely growing cells show enrichment of **neurodevelopmental signatures (blue/green)**." The underlined segments are confusing because they refer to a color key used in a different figure. Second, (blue/green) is meant to refer to diffusely growing cells, not neurodevelopmental signatures, right? Also, there are no true blue legends (deep green, teal, and turquoise). I mention this because the gradient actually used in Fig 2C is red/blue. Additionally, injury response appears partially enriched in both signatures. I believe this figure would improve significantly to support this claim if it was displayed separately for each line or grouped by cell types to save space and avoid confusing legends.
2. While I appreciate the authors' attempt to address my comment regarding Fig 3A, I simply meant to reduce the size of the font by half a point. Grouping two genes per row is slightly unorthodox and may be confusing to readers.

Version 2:

Reviewer comments:

Reviewer #3

(Remarks to the Author)

The authors have adequately addressed my comments.

One last minor comment:

It may be valuable to clarify that the "protein" values used are not protein expression levels (or signal intensity), but the fraction of cells with positive expression of this protein.

RESPONSE TO THE REVIEWERS' COMMENTS

Reviewer #1

In this study, Doroszko et al use PDX models and computational approaches to examine the potential links between tumor cell state and invasion routes. They show that cell lines enriched for MES-like and OPC-like state preferentially form dense tumours with perivascular invasion patterns, whereas lines enriched for NPC-like and AC-like fates are less dense and invade more diffusely. They then identify regulatory modules for each phenotype coalescing around ANXA1 for perivascularly invading tumors and HOPX/RFX4 for diffuse tumors. These are then functionally tested by CRIPR/Cas9 KO revealing a shift towards diffuse invasion in ANXA1-KO tumours and changes in fate, infiltration and proliferation in RFX4-KO tumors. The authors also show that KO of all three regulators extends survival.

This is overall an interesting paper, which is well written and executed. The computational approaches are a key strength of the study and reveal interesting links between cell fate and invasion patterns, that represent an important finding for the field. There are however some weaknesses, particularly in the functional studies, which need to be addressed in a revised manuscript to better support the authors conclusions.

Response. Thank you for the kind words and for the systematic assessment of the paper. We have now addressed the comments below, and added new time-lapse observations, interventions and analysis to strengthen the conclusions.

1. It is not appropriate to refer to unpublished work in demonstrating the two invasion patterns. This is one of the main findings of this study and the data should be shown, including the survival curves to better support the correlation between invasion modes and disease severity.

Response. We appreciate the reviewer's suggestion to clearly document the representativeness of the six selected GBM cultures. To address this point, we have now explicitly referenced our larger GBM resource consisting of 64 patient-derived cell line xenograft (PDCX) cultures recently made available on bioRxiv (<https://www.biorxiv.org/content/10.1101/2025.03.25.645260v1>). These 64 cultures have been thoroughly characterized for multiple invasion phenotypes, growth patterns, and survival outcomes, providing a robust dataset for assessing GBM heterogeneity. Principal component analysis (PCA) of these 64 PDCX cultures revealed two major invasion patterns: consolidated tumors with perivascular invasion and diffuse infiltration patterns. The six GBM lines used in our current study were selected based on their clear, stable, and reproducible representation of these patterns. To ensure that the HGCC cultures selected for this project matches those in our clinical collection, we performed an independent confirmation, obtaining phenotypic concordance scores of 88%-96% (c.f. Supplementary table 1). We

have now included an additional supplementary figure (Supplementary figure 1) clearly illustrating where these six models fall within the PCA distribution of the larger dataset.

Due to the extensive scope and the comprehensive nature of the larger dataset, we have documented this resource separately, as its detailed description and analysis extend beyond the focus of the current manuscript, which specifically investigates the transcriptional state underpinning GBM invasion phenotypes at single-cell resolution. We hope this clarifies our approach and sufficiently addresses the reviewer's concern.

Supplementary figure 1: PCA plot of phenotypic profiles of 64 PDCX models from the Human Glioma Cell Culture (HGCC) collection PCA map (<https://www.biorxiv.org/content/10.1101/2025.03.25.645260v1>). Each polygon represents a group of mice injected with cells from the same patients.

2. While the role of ANXA1 in driving the formation of dense lesions is well supported by the data, its specific role in perivascular invasion is not. Its expression is broad, the differences in expression in perivascular cells are not clear from Figure 4C. More evidence to support a direct role in invasion is required, staining of 'static' terminal tissue is not sufficient. As it stands the observed effect could also be a byproduct of the dense, mesenchymal phenotype and not due to a regulatory role for ANXA1 of invasive behavior. The authors should assess invasion in real time in the presence or absence of ANXA1 (e.g. in ex vivo systems) and consider conditional approaches for deleting ANXA1 at later stages of tumorigenesis to disentangle regulation of dense growth from invasion.

Response. Thank you for the comment. We have now added temporal observation data in three different experimental systems (co-culture, zebrafish xenografts, and mouse brain slice grafts) to evaluate the effect of ANXA1 knockout on not just condensation of cells and co-localization, but also movement along blood vessels. In a brain slice system, ANXA1 knockout U3013MG cells show significantly reduced motion along blood vessels compared to ANXA1 wildtype U3013MG cells, as determined of single-cell

tracking using time-lapse confocal data (Supplementary figure 12, Supplementary file 6 (video)). To further consolidate this phenotype, we carried out additional analyses with our zebrafish model (Almstedt et al, Neuro Oncology 2022), where we see a significant tendency for ANXA1 knockout U3013MG cells to spread diffusely, whereas wildtype cells co-localize with vessels, and also show collective movement along vessels (Supplementary figure 13 and Supplementary files 7 and 9 (videos)). We combined this with a third set of experiments in a co-culture system, where we also found that ANXA1-KO cells are less prone to engage with vessels (Supplementary figure 14). In addition to these data, we have added a histological analysis to document that U3013MG cells displace astrocytic end feet, which - albeit not a time-dependent observation - is a key characteristic of perivascular invasion (Watkins et al, Nature Communications 2014) (Supplementary figure 7). We have additionally performed an in vitro collagen sphere invasion assay comparing ANXA1-WT and ANXA1-KO U3013 lines. Although this does not prove its association with perivascular invasion, we saw a clear effect of cell invasion in the ANXA1KO line, consistent with a role in invasion (see Supplementary figure 11). Jointly the data support that condensed growths near blood vessels correspond to moving tumor cells.

3. To more convincingly demonstrate a role for ANXA1, gain-of-function experiments would also be required. Does expression of ANXA1 in diffuse lines switch them to dense/perivascular invading tumours?

Response. We appreciate this valuable suggestion and agree that gain-of-function experiments can strengthen our conclusions. We have now conducted additional experiments where we overexpressed ANXA1 in diffusely invading U3180MG cells, which express very low baseline levels of ANXA1. These modified cells demonstrated a clear shift in phenotype in zebrafish xenografts, becoming less diffuse and increasingly co-localizing and interacting closely with blood vessels (Supplementary figure 14, Supplementary files 8 and 10 (videos)). A similar phenotypic shift was confirmed in a complementary in vitro co-culture system, where ANXA1-overexpressing cells showed increased vessel association compared to controls (Supplementary figure 14). Furthermore, re-introducing ANXA1 into diffusely invading ANXA1-knockout U3013MG cells restored their original perivascular aggregation phenotype and interaction with blood vessels, emphasizing ANXA1's specificity in controlling this phenotype. Given the substantial duration required for mouse xenograft studies, we have not had time to confirm this gain-of-function phenotype shift in vivo in mice. However, the consistent results from two independent systems (zebrafish and in vitro co-culture) reinforce ANXA1 as a modulator of condensed growth and perivascular invasion. We believe these results sufficiently support our claim and substantially address the reviewer's valuable concern.

4. The results of the HOPX and RFX4 KO experiments are puzzling and do not fully support a key role for these genes in diffuse infiltration. Their effects seem multifactorial precluding a clear interpretation of the data. The increase in perivascular invading cells in HOPX-KO is not

convincing (Figure 6J). In addition, the reduced proliferation observed in both lines is a major confounder. The phenotypes observed may be simply due to reduced proliferation and altered differentiation and have little to do with invasion itself. Again, invasion should be assessed directly here and in real-time.

Response. Thank you for this insightful comment. We agree that the phenotypes resulting from HOPX and RFX4 knockout are more complex and multifactorial than those observed for ANXA1, and we appreciate the opportunity to clarify our interpretation. To further dissect the consequences of these knockouts, we have expanded our analyses in several key ways.

First, we performed in vivo single-cell RNA sequencing of HOPX-KO tumors, revealing a consistent shift in transcriptional state toward MES-like profiles, alongside reduced representation of AC-like and developmental gene signatures (Figure 7G–I). These results are complemented by projection of the differentially expressed genes onto the Eze et al. developmental brain atlas, where HOPX-KO cells showed enrichment for a neuroepithelial cluster defined by mesenchymal markers such as *ID3* and *ANXA2* (Supplementary figure 16). These data suggest that HOPX loss may relieve a brake on mesenchymal transdifferentiation.

Second, although the increase in perivascular-associated cells in HOPX-KO tumors (Figure 6J) was modest and may be interpreted as a trend, we further evaluated invasion-related behavior using a 3D co-culture system. In this assay, HOPX-KO cells showed a significantly higher tendency to interact with endothelial structures compared to controls (Supplementary figure 14), suggesting an increased affinity for vascular niches. While these results do not demonstrate perivascular invasion in the strict sense, they do support a functional link between HOPX loss and increased vessel association, which may represent a partial phenotypic shift rather than a complete route switch.

Third, to better contextualize the observed reduction in proliferation, particularly for RFX4-KO, we highlight that this is not merely a confounding variable but likely a reflection of altered differentiation state. Specifically, RFX4-KO led to decreased expression of astrocytic markers (e.g. *GFAP*), reduced *KI67* positivity (Figure 6L), and a transcriptional drift toward NPC/OPC-like signatures (Figure 7D–F). This supports a model in which RFX4 is not only involved in maintaining astrocytic-like invasion phenotypes, but also in sustaining proliferative capacity—a dual role that may explain the broader effects seen upon knockout. Notably, we also observed HOPX downregulation in the RFX4-KO, suggesting a potential regulatory connection between these factors (Supplementary Figure 15).

Altogether, while the phenotypic consequences of RFX4 and HOPX ablation are indeed more nuanced than those of ANXA1, our data support a model where these transcription factors function as key modulators of cell state within the diffuse invasive niche, influencing not only differentiation and proliferation, but also the cellular interaction with the brain microenvironment. We have clarified these

points in the Results and Discussion sections, and rephrased claims to better reflect the complexity of the observed phenotypes.

5. The GFAP results are also confusing as the authors show in Figure 7D/E that there is no change in the proportion of AC-like cells, yet they then show a decrease in AC-like signatures and GFAP staining in these tumours. Is AC fate really changing and is GFAP a robust readout for this? In general, the experiments relating to HOPX and RFX4 inactivation are too underdeveloped in their current form.

Response. We thank the reviewer for drawing our attention to these results, and agree that the analyses are pointing in somewhat different directions regarding AC-like differentiation. We went through these analyses carefully, and realised that in the data underlying Figure 7D/E both cells from culture and xenograft sources were included, instead of just xenograft as for ANXA1-KO in A/B. We already know from Figure 2 that the cell state distribution changes when cells are exposed to conditions in the mouse brain, where cells seem to be able to take on a wider variety of cell states. When only including xenograft cells in 7D/E, there is a clear decrease in AC-like proportions in the RFX4-KO, aligning with the pathway analysis and GFAP stainings. This effect was masked when cells from culture conditions were included. We have now changed the visualization from mosaic plots to stacked bar plots, as suggested by another reviewer, which in a clearer way demonstrate the decrease in AC-like cells upon RFX4-KO. To summarize, when RFX4 is knocked out in U3180MG cells the cells become less AC-like, as demonstrated through quantification of number of AC-like cells, decrease in AC-like signature as well as through GFAP stainings (which has now been moved to the supplementary figure 15).

Minor points

Sentence: “these latter were also enriched” on page 5 is confusing and should be rephrased.

Response. The sentence is now changed to a more clear statement.

It would be helpful to label the PDX models with their migration pattern in addition to the U number in the text and figures.

Response. We have now added the corresponding migration patterns to figure 2 and hope that it makes the results easier to follow.

The clarity of Figure 2C could be improved.

Response. Figure 2C displays the clusters derived from the same cells that are on the UMAPs from 2A and 2B. We show the enrichment of different gene signatures related to neuro-oncology and neural development. From our analysis, we see that the perivascular invading cells generally enrich for injury response signatures while diffuse growing cells show positive enrichment of neuro-developmental signatures. We have elaborated further on what is shown in figure 2 legends.

The difference in survival shown in Figure 5b are not convincing. It would be more appropriate to move this panel to a supplementary figure.

Response. Thank you for the comment. With the suggestion of reviewer 3, we have reanalyzed the survival data from patient samples in figure 5, assessing the survival differences using cox regression with age and sex as covariates. From the result of this analysis, it is clear that only the expression of RFX4 is significantly associated with survival in patients, and we have therefore chosen to only keep the RFX4 Kaplan-Meier plot in figure 5, together with the results from the multivariate survival analysis. Additional data has been moved to the supplement.

Reviewer #2 (Remarks to the Author): expert in bioinformatics and spatial proteomics

Doroszko et al. present an interesting paper on the invasion phenotypes of glioblastoma. I must confess this biological areas is quite distinct from my own but I am familiar with the computational and experimental techniques so I won't comment on the usefulness of the results to the community. I appreciate through the careful references to controls and potential confounding aspects of the analysis.

The approach looks sound and generally good care is taken throughout the manuscript to explain the intent and logical flow. The place where the manuscript could use most improving is the methods section where I am left guessing in many place what the others actually did. I suspect they are sensible things but I wouldn't be able to reproduce the analysis myself from this description. A list of uncertainties I had are below but I think the authors could apply a general principle of being more precise with the methodology descriptions:

Response: Thank you for these insightful comments on improving the clarity in how we communicate our analyses. We have worked through the Methods to improve the reproducibility of our analysis, with a specific focus given to the below points (see answers below).

- 1. Which highly expressed genes were removed from the analysis (currently vague)?*

Response: The following highly expressed genes were filtered out: *MALAT1*, mitochondrial genes (through command `! grep("^MT-")`), ribosomal genes (`! grep("^RP[SL]")`) and hemoglobin genes (`! grep("^HB[AB]")`). The downstream analysis was carried out with and without these filters applied without observing major changes of the results. In the end, since it significantly decreased object size, we used the filtered object.

2. What is cell-cycle regression?

Response: Cell-cycle regression is a computational technique to mitigate the effects of cells being in different phases of the cell cycle and therefore appearing more different in the transcriptional space than they are. The authors of the Seurat package (Butler et al., 2018) have developed two alternate workflows for cell-cycle regression, either completely removing the cell-cycle signal, or only removing the difference between cells in G2/M or S-phase, but retaining information on cells being actively cycling (S or G2/M) versus non-cycling (G1). In our case, we've employed the second workflow, since information on cells being cycling or not is an important biological signal (e.g. whether they are more stem-like or differentiated), while information on whether a cell is in S or G2/M-phase is not. We've added details in Methods to explain how this was done.

3. Louvain with multilevel refinement -> why was this chosen over other options?

Response: Louvain with multilevel refinement is more locally accurate. We reasoned that since our data contained quite divergent groups that have been merged into a community structure, optimizing it locally could have benefits. However, this way of clustering didn't produce much different separation than other algorithms, and we additionally want to point out that the clustering was mainly used for visualization purposes, specifically for the gene set enrichment analysis in Figure 2C, and therefore didn't impact downstream analyses.

4. Range of resolutions from 0.01 to 1 what was actual values tried?

Response: The following resolutions were tested: 0.01, 0.03, 0.1, 0.2, 0.3, 0.4, 0.5, 0.7, 1. Using Clustree, we generated a clustering tree (below) and through visual inspection determined that resolution 0.3 was most suitable and was therefore used for continued cell subpopulation identification.

5. *How was the dendrogram computed (e.g. average linkage?)*

Response. We realize that our original phrasing may have caused some confusion. To clarify: Louvain clustering, which we used to identify cell communities, is a graph-based method and does not produce a dendrogram in the traditional hierarchical clustering sense. Instead, it optimizes modularity on a nearest-neighbor graph to assign cells into communities.

The “dendrogram” we referred to was the directed acyclic graph (DAG) produced by the Clustree package, which visualizes how clusters merge or split across resolutions — but it is not a dendrogram derived from hierarchical clustering. We apologize for the ambiguous wording and have revised the text for clarity in the updated Methods section.

6. *"we assessed the stable and therefore optimal cluster separation" - There are two problem here I don't know what you actually did and I don't know how the second part of the sentence follows*

We have based our clustering decisions on the output made by Clustree as explained above in comment 4.

7. *Range of penalization value - again would be helpful to be precise*

Response: We tested the values 0.01, 0.05, 0.1 and 0.5. We've added this information in Methods in the main manuscript.

8. *What was the actual penalization parameter chosen and what were the values of the metrics.*

Response: The penalty parameter was chosen to 0.1, this information is now included in Methods. To determine the optimal penalty parameter, two metrics were developed in (Larsson, Held, et al. Nature Communications, 2024), which work in a similar manner to a silhouette score and the goal is to find the joint change point when plotting the values of the metrics for each penalty parameter tested. The penalty parameter where this change point occurs is the optimal one. For more details on these metrics we refer to \cite{pmid39516198}.

9. *Heirachical cluster settings missing again.*

Response: Thank you, we have added the details missing in Methods in the main manuscript.

The figures were generally hard to read if they contained numeric of textual detail. Whilst many could use improvements those in figure 7 I could not really deduce what I was meant to read.

Response: Thank you for this comment. We have worked through our figures to make them easier to read. To improve the clarity of figure 7 we have replaced the mosaic plots with stacked bar plots.

"Our initial scRNA-seq analysis revealed a significant corrleation between ..." - at the moment this is simply a claim with no evidence, a figure or some data is missing I think.

Response: Thank you for this comment. We assume that the reviewer refers to the statement that our data revealed a significant correlation between transcriptional cell states and *in vivo* invasion routes. This statement is based on the analysis underlying figure 2F, where it's evident that certain cell states are more abundant in samples displaying a specific invasion pattern (e.g. the AC-like state and diffuse invasion). This relationship is statistically assessed using a chi-squared test, with a p-value of $\lll 0.0001$ and visualized as a mosaic plot in figure 2F. To clarify where the evidence for this statement can be found, we have included a reference to figure 2F after the mentioned statement in the main manuscript.

Reviewer #3

Doroszko et al. investigate the invasion phenotypes of glioblastoma (GBM) and how they are influenced by plastic and reprogrammable cell states. The study characterizes the invasion

phenotypes of 64 GFP/luciferase-tagged HGCC cultures in nude mice (although this data is not provided), identifying two main phenotypes: consolidated tumors with perivascular invasion and diffuse growth patterns. Six cultures were selected for further analysis based on their consistent phenotypic expression. The authors use scRegClust to identify regulators associated with invasion routes, with specific genes linked to perivascular (e.g., ANXA1, ANXA2) and diffuse (e.g., HOPX, CKB) invasion. The findings suggest that specific transcriptional states correlate with distinct invasion patterns in GBM, with ANXA1 identified as a key regulator of perivascular invasion.

Major comments

1. The authors do not discuss the relevant clinical information from the patients and their consistency or disagreement with their experimental findings. For example:

a. Example 1: The survival patterns of both groups in host mice do not correlate or agree with the actual survival of the patients according to Fig 1 E and D. The entire purpose of modeling tumors in mice is to produce conclusions that help us predict the behavior of these tumors in human patients, but in this case, that isn't true and isn't discussed. There are many likely reasons for this, such as tumor:immune interactions. Since the mouse modeling was performed in immune-compromised mice, lacking an essential component of the tumor microenvironment, conclusions from these experiments should be carefully evaluated and thoroughly discussed.

Response. Thank you for this comment. We agree that mouse grafting models do not capture all aspects of human disease. Our panel does reproduce a range of invasion behaviors also observed in humans which make them an interesting and useful model to pursue mechanisms of variable growth morphology. Key differences include that patients receive therapeutic surgery, radiotherapy, and can produce a degree of adaptive immune responses against the tumor; these aspects are not included in the model. The absence of surgical treatment of the mice helps explain their shorter survival. Tumors that form a solid bulk are more severe for the mice once it reaches a certain size due to the increased pressure in the skull. We also observe leptomeningeal invasion more often in these cell lines which can cause the blockage of the cerebrospinal fluid and decrease mice survival. In contrast, the diffuse growing lines spread to the whole brain easily and therefore does not cause terminal pressure. This is now discussed more fully in the discussion section. In the original submission, we did extend to an analysis of patient samples from Brain UK, confirming an association between perivascular invasion and ANXA1, and diffuse invasion and HOPX, and were thus evaluated. RFX expression was hard to assess in the BrainUK but was associated with survival in our cohort, independently of sex and age (below). Future studies will be required to disentangle the specific contribution of different modes of invasion in patients, and for this our markers provide an important step forward.

b. Example 2: Fig 1 E (patient clinical information) shows that all 3 PDCs with diffuse invasion phenotype are derived from patients with Classical subtype tumors, while the perivascular PDCs come from Mesenchymal and Proneural tumors. Previous reports (Neffel et al.) showed that while each patient tumor is composed of a mixture of cell states, proneural subtype tumors are mainly composed of NPC-like and OPC-like cells, Mesenchymal subtype tumors are mainly composed of Mes-like cells, and classical tumors contain more AC-like cells. While this somewhat agrees with Figure 2D,E, it was not discussed. Additionally, stacked bar plots showing the distribution of each subtype per sample would be a valuable addition to show the exact composition of each sample since butterfly plots can be tricky to interpret, especially when cells are close to the axis.

Response. Thank you for pointing this out. The subtype assignments in figure 1 are based on previously published bulk profiling and the bulk signatures by Wang et al. We agree that these signatures capture very similar aspects as those by e.g. Neffel et al. To make the presentation fully clear for all readers, we have added the stacked bar charts of relative abundance of each state for the cell lines in supplementary figure 4. We thank the reviewer for this suggestion and we do emphasize that the main question here is whether these signatures relate to invasion routes.

c. Are there any H&E from the original 64 (or at least the 6 used in the paper) patients provided by the HGCC that confirm this phenotype in humans?

Response. We have tumor samples from 4 patients out of the 6 used in this paper. We have added the H&E stainings now as a supplementary figure 8. Unfortunately all the tissue samples we have from the HGCC are from the tumor core which makes it challenging to assess invasion patterns at the invasive front. This is why we have included the Brain UK cohort where we could analyze both the tumor core and the invasive edge (Figure 5), also showing statistics on the Pathologist's assessment to document the differential presence of ANXA1, HOPX, and RFX4 by invasion mode.

2. The authors claim that the dimension reduction of the scRNAseq data revealed “distinct regions” for the cells derived from the two classes. However, the method used for integration (RPCA) has been benchmarked and shown to produce poor results. That being said, did the authors consider any other integration approaches? Was the sex of the patient (two males and four females) accounted for? Multiple batches? Etc.

Response. Thank you for this important point. We acknowledge that RPCA-based batch integration may have limitations and appreciate the opportunity to clarify our approach. In our study, RPCA followed by UMAP was used solely for visualization in figure 2A–C, to highlight broad trends across all samples. However, our core downstream analyses, including transcriptional state mapping (Figure 2D–F) and

regulatory inference using scregclust (Figure 3), do not depend on RPCA or any specific integration method. Instead, we adopt a late integration strategy, similar to what has been described in e.g. Neftel et al. and other glioma studies, where gene modules are learned from individual samples and compared across conditions. Importantly, the observed increase in transcriptional diversity among in vivo cells (Figure 2B) is not an artifact of RPCA. This increased heterogeneity was also evident in the cell state assignment analysis (Figure 2D, F), which uses independently assigned cell states based on Neftel et al.'s framework and does not involve batch correction or RPCA. Regarding potential confounding by sex, we note that sex was not included as a covariate in the RPCA alignment, as this integration was not used for statistical modeling. However, since the observed state shifts (e.g., the transition to astrocytic states in ANXA1-KO) are supported by multiple independent analyses — including differential gene expression, spatial proteomics, and enrichment against developmental atlases — we believe the findings are robust to batch and patient-level effects. To further ensure transparency,, we now clearly indicate the role of each integration step in our revised Methods and Results, and have softened any language suggesting causal interpretation from the RPCA+UMAP projection alone.

3. The authors claim that “U3220MG displayed a specific cluster apparently associated with its high degree of leptomeningeal invasion (Figure 2A).” The wording of this phrase is slightly confusing. How was the association determined? Was there any molecular/marker analysis done to back up this statement? Is this observation consistent through the 2 PDX samples and 1 tissue culture sample from this line?

Response. Thank you for pointing this out. We agree that the original phrasing was ambiguous and may have suggested a causal or exclusive link between the U3220MG-specific cluster and leptomeningeal invasion. To clarify, we have revised the sentence in the Results section to read:

“Notably, U3220MG, which displays a high degree of leptomeningeal invasion (Figure 1A, C), also harbored a distinct transcriptional cluster (Figure 2A), suggestive of a unique cell state potentially linked to this invasion route.”

This revision avoids overinterpretation while still highlighting a potentially interesting observation.

4. Some experimental details are confusing/missing. For example, while the authors mention that they used 19 samples for the scRNAseq in Fig 2A, they do not mention how many samples per group (12 groups (6 cell lines and 2 conditions)). The deposited data in GEO show only 16 samples that are mentioned in this figure and 4 additional samples with ANXA1KO and RFX4KO (not 19).

Response. Thank you for checking this. The GEO submission had 3 samples missing, which has now been addressed.

5. How was the “shorter list of promising regulators” compiled? What were the cutoffs or criteria used?

Response. We thank the reviewer for this helpful question. To compile the shorter list of regulators for experimental validation, we applied a combination of statistical and practical criteria. First, we prioritized genes that were significantly associated with invasion route (perivascular, diffuse, or leptomeningeal), based on ANOVA and differential expression analyses (adjusted p-value < 0.01, absolute log₂ fold change > 0.5). We excluded regulators that were predominantly associated with patient identity or growth condition, as shown in figure 3C–E, to avoid confounding effects. From the resulting list, we selected regulators with established or plausible roles in invasion, differentiation, or transcriptional control, and for which high-quality antibodies were available for spatial protein validation. This process led us to focus on ANXA1, RFX4, and HOPX, which emerged as strong and feasible candidates. These steps are now clarified in the revised Methods section (Section 8).

6. Were multivariate survival analyses performed accounting for age/subtype/sex and mutations? Since these markers were associated with different growth phenotypes, which supposedly have different survival implications, it does not make sense that all markers predict worse survival in patients. Additionally since the growth patterns of these PDXs have been assessed, the survival should be conducted for each growth pattern separately.

Response. In response to Reviewer 3’s suggestion, we reanalyzed the survival data from the HGCC patient cohort using multivariate Cox regression, adjusting for age and sex. This analysis revealed that high RFX4 protein expression is significantly associated with shorter patient survival (HR = 1.014, p << 0.001), while ANXA1 showed a modest but statistically significant association (HR = 1.003, p < 0.05). HOPX expression did not show a significant relationship with survival in the multivariate setting.. Based on these findings, we retained the RFX4 Kaplan-Meier plot in the main Figure 5 and moved the original analyses to the supplementary material (Supplementary figure 8). We also now provide the underlying quantification as a supplementary data file.

Minor comments:

1- Figure 3 labels/gene names are too close to eachother and hard to read

Response. Thank you for the suggestion. The figure 3 labels are now fixed for clarity.

2- Legends are missing important information that is needed for the reader to understand the figures for examples:

a. Figure 1E, what is the unit of survival (days?). Also abbreviations/locations should be defined.

Response. We have updated the legends for figure 1E to clarify all abbreviations.

3- Some such information is present in the method but should be included in the legends for examples:

a. In Figure 5B, the type of “expression” should be clearly annotated protein IHC vs transcriptomics. The high/low threshold should be mentioned in the legends. It would also be beneficial if the expression scores per patient for all the markers assessed to be provided for all 148 patients alongside their clinical information. This information should be provided in the legends. This is especially important since the HGCC resource that’s available online only contains 48 samples as opposed to the 148 samples. The online resource provides gene expression profiling data that is downloadable in z score format. When using this information to assess survival, neither ANXA1, HOPX or RFX4. Although ANXA1 and HOPX show the same trends. It’s worth mentioning that in the TCGA GBM dataset neither of these markers are prognostic. When using the TCGA GBMLGG combined dataset, the markers are prognostic but it’s only suggestive of association with Glioblastoma not low-grade gliomas. This may be a distinction between protein vs RNA expression but needs to be discussed.

Response. Thank you for the comment. We have now added to the discussion that while we see associations for RFX protein in our cohort using Kaplan Meiere analysis and multivariate Cox regression, the Affymetrix bulk RNA profiling scores (Xie et. al. 2015) in the HGCC data sets are not prognostic. We did look up the RNA levels for all genes in the OSgbm resource, which integrates survival data for seven different cohorts and find a significance for ANXA1 RNA as associated with shorter survival in GBM patients (plot below), but not the others. This observation is also mentioned in the discussion. We believe that accounting for survival trends in our cohort can be of interest to readers, and we hope that the revised version strikes a balance. The TMA protein scores and clinical data have been uploaded as a supplementary table (Supplementary file 11).

[REDACTED]

4- PDXs are serially transplanted from mouse to mouse and not cultured first. The point of PDXs is to mimic the in vivo component and since these cells were handled in vitro prior to transplantation (cultured , genetically engineered to express GFP-Luciferase), I believe they're no longer true PDXs and the distinction should be made (Mouse xenografts from PDCs rather than PDXs).

Response. Thank you for the comment, we have now updated the terminology to reflect the difference, consistently using the term patient derived cell-culture xenografts, PDCXs.

5- Supplementary table with differential expression information between clusters/samples from the scRNAseq should be provided.

Done. This is now provided as supplementary file 5.

REVIEWER COMMENTS

Reviewer #1 (Remarks to the Author):

The authors addressed most of my comments, particularly regarding the functional studies on ANXA1. The manuscript has improved greatly overall, though the mechanistic work on RFX4 and HOPX remains underdeveloped. It would be important to tone down the statements around ANXA1 playing a specific role in perivascular invasion in the final manuscript. The presented data only support a role for ANXA1 in vascular association and formation of a bulk, but no conclusive evidence on perivascular invasion is provided. The conclusions are therefore overstated in their current form.

Response: Thank you for the kind words regarding the progress of the manuscript and for highlighting this important point. Carefully re-reading the text, we agree that our original phrasing overstated the mechanistic evidence regarding ANXA1's direct role in perivascular invasion. We have adjusted the manuscript to clarify that, although our dynamic analyses clearly associate ANXA1 with perivascular growth patterns and vascular association, its exact role in these processes is not fully resolved. Specific revised statements throughout the manuscript reflect this more cautious interpretation (see below for details).

- **Abstract, original statement:**

"Computational modeling identifies ANXA1 as a driver of perivascular invasion in GBM cells with mesenchymal differentiation..."

Revised statement (changes in red):

"Computational modeling identifies ANXA1 as a driver of perivascular **involvement** in GBM cells with mesenchymal differentiation..."

- **Results, original statement:**

In summary, the absence of ANXA1 in tumor cells reduced the density of the tumor bulk, eliminated perivascular invasion and instead shifted the main invasion mode to diffuse invasion.

Revised statement (changes in red):

In summary, the absence of ANXA1 in tumor cells reduced **tumor bulk formation and significantly reduced association with vascular structures, with tumor cells shifting toward a more diffusely infiltrative phenotype.**

- **Results, original statement:**

"Taken together, these complementary dynamic analyses robustly reinforce ANXA1 as a critical mediator of perivascular invasion dynamics in GBM cells."

Revised statement (changes in red):

"Taken together, these complementary dynamic analyses support a role for ANXA1 **in promoting dynamic tumor cell association with blood vessels.**"

- **Original statement:**

Overall, these findings underscore the heterogeneity of protein expression in invasive GBM and provide further support for ANXA1 protein as a selective marker of perivascular invasion in GBM and HOPX and RFX4 as candidate protein markers for diffuse route-invading GBM.

Revised statement (changes in red):

"These findings underscore the heterogeneity of protein expression in GBM and further support ANXA1 protein **as a marker associated with perivascular localization and dense growth patterns**, and HOPX and RFX4 as candidate protein markers for diffuse route-invading GBM.

- **Discussion, original statement:**

"ANXA1 emerged as a pivotal gene in GBM perivascular invasion. Knocking out ANXA1 in perivascular invading cells induced notable phenotypic shifts, including the loss of tumor bulk and perivascular invasion..."

Revised statement (changes in red):

"In this study, ANXA1 emerged **as strongly associated with perivascular growth patterns in GBM**. Knocking out ANXA1 induced phenotypic shifts characterized by reduced tumor bulk, decreased and vascular involvement, while acquiring an AC-like cell state and diffuse invasion, ultimately leading to increased median survival in mice. "

Reviewer #2 (Remarks to the Author):

The reviews have responded well to by comments. I have no further comments.

Response: Thank you for the careful assessment of our manuscript.

Reviewer #3 (Remarks to the Author):

Overall, the authors have made significant efforts to address the reviewers' concerns and provide additional data to support their conclusions. The manuscript is now more robust and comprehensive, but some comments require further attention.

Points Not Addressed:

1. In my original comments (#6), I stated: "Were multivariate survival analyses performed accounting for age/subtype/sex and mutations? Since these markers were associated with different growth phenotypes, which supposedly have different survival implications, it does not make sense that all markers predict worse survival in patients. Additionally since the growth patterns of these PDXs have been assessed, the survival should be

conducted for each growth pattern separately." I appreciate the additional analyses performed by the authors; however, the multivariate analyses only account for sex and age and not subtype and mutations. Additionally, an important point was not addressed. Since the growth patterns of at least 64 of these PDXs are assessed by the authors and available, why not perform the survival analysis separately by growth pattern?

Response: We thank the reviewer for their insightful and important suggestions. We agree that including molecular subtype and mutation status alongside age and sex in the survival analysis provides a more complete view of the potential prognostic role of RFX4 protein. We have now extended our multivariate Cox regression models accordingly. Specifically:

- **Inclusion of Subtype:** We incorporated transcriptional subtype (Mesenchymal, Classical, and Proneural) as covariates in the Cox regression. This analysis revealed a modest association between the Mesenchymal subtype and overall survival, in line with findings from previous studies. Importantly, including subtype did not alter the statistical significance of RFX4 protein as a survival-associated factor ($p = 0.00856$). As a technical sidenote, we clarify that, for categorical data like subtype R automatically uses one of them (classical) as the baseline, while reporting p values for the other two.
- **Mutation and CNA Status:** We also explored whether key genomic alterations listed in Figure 1 (TP53, PTEN, PIK3CA mutations; CDKN2A, EGFR, CDK4 CNAs) affected the relationship between RFX4 expression and survival. Individually, none of these features correlated significantly with patient survival (all p-values > 0.3 , Cox regression). When included as covariates in the Cox model, together with sex, age, and subtype, they had limited impact on the association between RFX4 in all cases except CDK4. This may potentially point towards a role for CDK4, which would require larger materials and is reserved for future work.
- **Stratification by Growth Pattern:** As suggested, we conducted a subgroup survival analysis based on xenograft-derived invasion phenotypes (diffuse vs. perivascular). While the sample sizes were reduced, we observed that RFX4 protein expression remained significantly associated with shorter survival in the diffuse invasion group ($p = 0.0304$), highlighting a potentially route-specific prognostic role. Given the limitations in sample size, we suggest that future studies with larger and more diverse cohorts would be crucial to fully explore these findings.
- **Discussion Revision:** We have revised the results and discussion to more accurately reflect the scope and interpretation of the survival findings, and also list p-values for each protein (ANXA1, HOPX, RFX4) when doing different corrections and subselections (new Supplementary table 4). Added to discussion:

"In the present cohort, we noted an association between RFX4 protein expression and shorter survival in unselected GBM patients, also after correcting for age, sex, and transcriptional subtype. RFX4 was also associated with survival within the subgroup of patients with a diffuse growth phenotype in mice (Supplementary table 4). These findings may warrant validation in larger, independent patient cohorts."

We thank the reviewer again for this critical feedback, which has significantly improved the rigor of our survival analysis and the clarity of our conclusions.

2. In my original comments (#4), I stated: "Some experimental details are confusing/missing. For example, while the authors mention that they used 19 samples for the scRNAseq in Fig 2A, they do not mention how many samples per group (12 groups: 6 cell lines and 2 conditions). The deposited data in GEO show only 16 samples that are mentioned in this figure and 4 additional samples with ANXA1KO and RFX4KO (not 19)." While the authors kindly responded by adding the missing samples to the GEO repository, they did not clarify the statement in the manuscript regarding those 19 samples. These samples can be summarized into 12 groups (2 conditions and 6 lines) with some conditions having 2-4 replicates while others having a single replicate. I do not understand the need to explicitly state the number 19 without clarifying what is in those 19 samples. Was there a specific reason why 3054 PDXs required 4 replicates?

Response. It's a fair point that it is more informative to write down what the groups are, rather than the total sample count.

Old text: This encompassed samples from adherent cultures before injection and tumor cells isolated from mouse brains at experimental endpoint, totaling 19 scRNA-seq samples with 119,766 cell transcriptomes passing quality control.

Revised: This encompassed samples from adherent cultures before injection and tumor cells isolated from mouse brains at experimental endpoint. **The final data contains 119,766 cell transcriptomes, covering the six lines under *in vitro* and *in vivo* conditions, i.e. 12 groups (samples specified in Methods).**

There was no specific reason why U3054MG has more replicates, other than our preference to have multiple runs for one case.

3. In the rebuttal, the authors' response to comment number 5 is not reflected in the edits to the referenced methods in section 8.

Response. We apologize for this oversight. This information has been added to section 8 of methods.

*4. While the new legends include appropriate annotation of the expression used for survival (protein), the methods still state: "Survival analysis (multivariate). Cox proportional hazards regression was performed using median-dichotomized **gene** expression levels (e.g., HOPX, ANXA1, RFX4) as covariates alongside age and sex, with survival time as the outcome."*

Response. Thank you for noting this typo. We have now corrected to explain that protein, not median-dichotomized gene expression levels..

5. More importantly and adding to point #4. When I attempted to replicate the survival results, I was met with a couple of challenges. #1 survival status is not provided (1 or 0, I assumed that all are 1s for this test) #2 There are multiple points per patient in supplementary table #11 with no mention in the methods of how these replicates are accounted for. Specifically, 52 patients have 4 replicates, 8 patients have 3 replicates, 82 patients have 2 replicates, and 8 patients have a single replicate. I was able to generate an almost identical Kaplan Meier curve for RFX4 only when using all 394 data points with provided survival information but not when using a single (averaged) point per patient which is the correct analysis. This is extremely concerning and needs to be urgently addressed.

Response. We thank the reviewer for this critical and detailed feedback, which led us to strengthen our survival analysis considerably. Specific clarifications:

Clarification of Survival Data. As correctly noted, all patients in the cohort were deceased, meaning there are no censored observations. We have explicitly clarified this point in Methods section 12 to preclude misunderstanding.

- **Addressing Nested Design in Cox Models.** The reviewer rightly points out that the use of multiple TMA cores per patient introduces a nested data structure that must be properly accounted for in survival analyses (in the new submission, the TMA data is listed as supplementary table 6, instead of supplementary file 11). To address the issue of multiple replicates per patient, we implemented a shared-nesting Cox proportional hazards regression model, using the cluster() option in R to adjust for within-patient correlations (syntax given in Methods). This adjustment ensures that the standard errors correctly reflect the nested structure of the data, where multiple samples from the same patient may be more similar to each other than to samples from other patients. Given that most patients have only 1–2 replicate cores (with only a few having 3 or 4), we opted for cluster() rather than frailty(), which typically requires a higher number of observations per group.
- **Comparison to Naive Averaging Approach.** For completeness, we also performed a Cox regression using a naive approach where RFX4 protein levels were averaged per patient. When using the naive averaging approach and adjusting for age, sex, and subtype, the association between RFX4 protein levels and survival remained statistically significant ($p = 0.0349$).
- **Kaplan-Meier Re-analysis.** We further revisited our Kaplan-Meier analysis using patient-level averaged protein values and observed that while median-dichotomized expression yielded a borderline separation, stratification into tertiles (top vs bottom third) revealed a clearer survival difference, below. Further analysis of RFX4 protein in bigger cohorts would be required to conclusively settle this, reserved for future work.

- **Improved Methods description.** All details of the updated statistical approach, including model choice and justification, have now been incorporated into Methods section 11.

We greatly appreciate the reviewer’s detailed feedback, which prompted us to revisit our survival analysis. These refinements have strengthened the rigor of our approach and clarified the interpretation of our results within the broader context of the study.

Minor Comments:

1. The legend for Fig 2C is a little confusing. It states: "UMAP of GBM cells displaying enrichment of different gene signatures of the clusters. NES refers to normalized enrichment score. Perivascular invading cells (yellow/brown) exhibit enrichment for injury response, oligodendrocyte, and macrophage signatures, while diffusely growing cells show enrichment of neurodevelopmental signatures (blue/green)." The underlined segments are confusing because they refer to a color key used in a different figure. Second, (blue/green) is meant to refer to diffusely growing cells, not neurodevelopmental signatures, right? Also, there are no true blue legends (deep green, teal, and turquoise). I mention this because the gradient actually used in Fig 2C is red/blue. Additionally, injury response appears partially enriched in both signatures. I believe this figure would improve significantly to support this claim if it was displayed separately for each line or grouped by cell types to save space and avoid confusing legends.

Response. Thank you for catching this detail. We have now re-written the legend for clarity and to avoid overstatement. We appreciate the suggestion to show cells separately, but this was hard to fit within the figure. The legend for 2C now says: "(C) UMAP of the same GBM cells as

in A and B, displaying enrichment of different gene signatures, measured by the Normalized Enrichment Score (NES) in each cell cluster. Note the differential distribution of injury response, oligodendrocyte, and macrophage signatures, versus neurodevelopmental signatures."

We think that the present figure and this new legend clearly conveys the key points and will be accessible to readers.

2. While I appreciate the authors' attempt to address my comment regarding Fig 3A, I simply meant to reduce the size of the font by half a point. Grouping two genes per row is slightly unorthodox and may be confusing to readers.

Response. Done.

REVIEWERS' COMMENTS

Reviewer #3

The authors have adequately addressed my comments. One last minor comment: *It may be valuable to clarify that the "protein" values used are not protein expression levels (or signal intensity), but the fraction of cells with positive expression of this protein.*

Response: We have now made this clarification.

Using all 394 points

Using single point per patient

From the manuscript